# Dissemination of carbapenemase-producing Enterobacterales in the community of Rawalpindi, Pakistan

Amina Habib[1☉], Stéphane Lo[2,3☉], Khanh Villageois-Tran[2,4], Marie Petitjean[2], Shaheen Akhtar Malik[5], Laurence Armand-Lefèvre[2,3], Etienne Ruppé[2,3‡], Rabaab Zahra[1‡]*

1 Department of Microbiology, Faculty of Biological Sciences, Quaid-i-Azam University, Islamabad, Pakistan, 2 Université de Paris, IAME, INSERM, Paris, France, 3 APHP, Laboratoire de Bactériologie, Hôpital Bichat, Paris, France, 4 APHP, Laboratoire de Bactériologie, Hôpital Beaujon, Clichy, France, 5 Accident and Emergency Department, Benazir Bhutto Hospital, Rawalpindi, Pakistan

☉ These authors contributed equally to this work.
‡ ER and RZ also contributed equally to this work as co-last authors.
* rzahra@qau.edu.pk

**Data Availability Statement:** The data relevant to this study are available from the National Centre for Biotechnology (NIH) at accession number PRJNA645311 (https://www.ncbi.nlm.nih.gov/bioproject/PRJNA645311).

## Abstract

Carbapenems are considered last-line beta-lactams for the treatment of infections caused by multidrug-resistant Gram-negative bacteria. However, their activity is compromised by the rising prevalence of carbapenemase-producing Enterobacterales (CPE), which are especially marked in the Indian subcontinent. In Pakistan, previous reports have warned about the possible spread of CPE in the community, but data are still partial. This study was carried out to analyse the prevalence of CPE, the genetic characterisation, and phylogenetic links among the spreading CPE in the community. In this cohort study, we collected 306 rectal swabs from patients visiting Benazir Bhutto hospital, Rawalpindi. CPEs were screened by using ertapenem-supplemented MacConkey agar. Identification was performed by using conventional biochemical tests, and genomes were sequenced using Illumina chemistry. Antibiotic resistance genes, plasmid incompatibility groups, and *Escherichia coli* phylogroups were determined in silico. Sequence types were determined by using MLST tool. The prevalence of CPE carriage observed was 14.4% (44/306 samples). The most common carbapenemase-encoding gene was $bla_{NDM-5}$ (n = 58) followed by $bla_{NDM-1}$ (n = 7), $bla_{NDM}$ (non-assigned variant, n = 4), $bla_{OXA-181}$ (n = 3), $bla_{OXA-232}$ (n = 3) and $bla_{NDM-7}$ (n = 1). Most of the CPE were *E. coli* (55/64, 86%), and the genomic analysis revealed a pauciclonal diffusion of *E. coli* with ST167 (n = 14), 405 (n = 10), 940 (n = 8), 648 (n = 6) and 617 (n = 5). We obtained a second sample from 94 patients during their hospital stay in whom carriage was negative at admission and found that 7 (7.4%) acquired a CPE. Our results indicate that the prevalence of CPE carriage in the Pakistani urban community was high and driven by the dissemination of some *E. coli* clones, with ST167 being the most frequent. The high CPE carriage in the community poses a serious public health threat and calls for implementation of adequate preventive measures.

**Funding:** This work was partially supported by the "Fondation pour la Recherche Médicale" (Equipe FRM 2016, grant 325 number DEQ20161136698); the Direction Générale des Armées (project FastGeneII); the Programme Hubert Curien "Peridot" (Ministère des Affaires Etrangères, France) and by the Higher Education Commission, Pakistan under Pak-France Joint Grant (PERIDOT) to RZ and ER, National Research Program for Universities (NRPU - Project 9755) to RZ, International Research Support Initiative Program (IRSIP) (PIN: 43 BMS 16) to AH.

**Competing interests:** The authors have declared that no competing interests exist.

## Introduction

Enterobacterales reside as commensals in gut microbiota at concentrations of approximately $10^8$ colony-forming units (CFU) per gram of faeces [1]. However, they may act as opportunistic pathogens frequently in community and hospital-acquired infections [2], causing meningitis, septicemia, pneumonia, pyelonephritis, peritonitis, and device-related infections [3, 4]. Some important virulence factors involved in pathogenicity of members of Enterobacterales are enterotoxins, hemolysins, toxins, fimbriae, mannose binding type1-H adhesion, intimin, alkaline phosphatase, haemagglutinin, siderophores lipopolysaccharide (LPS), capsular polysaccharide and biofilm production [5–7]. Virulence factors support bacteria to adhere, invade and damage host cells, escape host defense mechanisms and cause infection [8].

Resistance to antibiotics in Enterobacterales has been on the rise since the past two decades due to the large diffusion of CTX-M-type extended spectrum beta-lactamases (ESBL) which confer resistance to most β-lactams except for carbapenems [9]. Accordingly, the latter are commonly used as last line antibiotics for treating infections caused by multidrug resistant Enterobacterales including those producing ESBLs. Hence, the use of carbapenems has grown in parallel to the rise of ESBL-producing Enterobacterales, paving the emergence and dissemination of carbapenem resistance. While Enterobacterales can resist to carbapenems by combining the production of ESBL and decreased outer-membrane permeability, the main mechanism for carbapenem resistance is the production of carbapenemases [10, 11]. The major carbapenemases in Enterobacterales belong to three classes, Ambler class A (e.g., *Klebsiella pneumoniae* carbapenemase, KPC), class B (e.g. New Delhi metallo β-lactamase, NDM) and class D (e.g. OXA-48 and its close relatives such as OXA-181) [12]. Among them, NDM is one of the most commonly identified carbapenemase worldwide and many studies have reported its high prevalence in the Indian subcontinent [13–15]. Moreover, carbapenemase-encoding genes are carried on plasmids which often harbour multiple other antibiotic resistance genes (ARGs), narrowing the range of treatments possibly active on CPE [16].

*E. coli* and *Klebsiella pneumoniae* are the most frequent carriers of $bla_{NDM}$, with specific sequence types (ST) such as ST11, 14, 15, 147 for *K. pneumoniae* and ST167, 410, 617 for *E. coli*. Diverse variants of carbapenemases such as VIM, IMP, NDM, KPC, and GIM have also been reported in Enterobacterales across Pakistan [17–19]. Limited data are available regarding the epidemiology of CPE carriage in the community [20–22], but suggesting a marked dissemination of CPE in Pakistan, Our study aimed to fill these gaps in assessing the CPE carriage prevalence in a Pakistani urban community and the underlying genetic determinants responsible for emergence of carbapenem resistance.

## Methods

### Ethics

The study was approved by the Ethical Review Committee, Rawalpindi Medical University on September 23, 2017. Verbal informed consent was obtained from all participants before the collection of samples, and it was written in participant's record. Patients who refused to participate in the study were excluded. Participation was voluntary, and no minor was included in our study.

### Patients and samples

Between January 2018 and May 2019, patients admitted to traumatology and gynaecology units of Benazir Bhutto Hospital, Rawalpindi, Pakistan with no antibiotic exposure, hospitalisation, and travel history abroad during the 3-months preceding the admission were included.

Rectal swabs (Nuova, Canelli, Italy) were collected within 24 hours of admission. Patients in whom the first swab was negative for carbapenem-resistant Enterobacterales (CRE), a second swab was taken 72 hours after admission.

## Bacterial isolation and identification

We initially did experiments to set the optimum culture conditions to recover CPE with respect to the epidemiological specificities of Pakistani CPE. We cultured already characterised, 4 strains of *E. coli* (2 each ertapenem-sensitive and ertapenem-resistant) [23] on MacConkey agar (Oxoid, Basingstoke, England) with increasing ertapenem (Merck & Co., Whitehouse Station, USA) concentrations ranging from 0.125 mg/L to 5 mg/L. We finally set the ertapenem concentration at 0.5 mg/L as the optimum concentration at which only resistant bacteria were growing while sensitive were inhibited.

Rectal swabs were thoroughly mixed with 1 mL normal saline and 100 μL of each sample was plated onto 0.5 mg/L ertapenem–supplemented MacConkey agar and incubated overnight at 37˚C. Colonies with distinct morphotypes (with respect to the colour, size, and shape) were sub-cultured on MacConkey agar and initial identification was performed based on colony characteristics, Gram staining (Daejung Chemicals Ltd., Siheung, Korea), and conventional biochemical tests. All the Gram-negative short rods were further tested for Enterobacterales through a range of biochemical tests [24], using, oxidase (BDH Laboratory Supplies, Poole, UK), sulfide Indole Motility (SIM) medium (Liofilchem, Roseto, Italy), triple Sugar Iron (TSI) agar (Liofilchem, Roseto, Italy), methyl red-Voges-Proskauer (MR-VP) broth (Liofilchem, Roseto, Italy), nitrate broth (Liofilchem, Roseto, Italy), urea broth (Liofilchem, Roseto, Italy) and Simmon's citrate media (Liofilchem, Roseto, Italy) [25]. After identification, pure bacterial colonies were stored in charcoal swabs (Nuova, Canelli, Italy) and shipped at ambient temperature to the bacteriology laboratory of the Bichat-Claude Bernard University Hospital, Paris, France. There, the bacterial species identification was confirmed by matrix-assisted laser desorption/ionization time of flight mass spectrometry (MALDI-TOF MS, Bruker, Bremen, Germany).

## Antimicrobial susceptibility testing

Enterobacterales isolates were tested for their susceptibility to 16 antibiotics from different antimicrobial groups: Penicillins: ticarcillin (75 μg), temocillin (30 μg), β lactams-β-lactamase inhibitor combination: amoxicillin-clavulanic acid (30 ug), ticarcillin-clavulanic acid (85 μg), Cephalosporins: cefoxitin (30 μg), ceftazidime (10 μg), cefotaxime (5 μg), cefepime (30 μg), Monobactams: aztreonam (30 μg), Carbapenems: ertapenem (10 μg), imipenem (10 μg), Aminoglycosides: amikacin (30 μg), gentamicin (10 μg), Quinolones: nalidixic acid (30 μg), ofloxacin (30 μg) and Sulfonamides: trimethoprim /sulfamethoxazole (30 μg) using the disk diffusion method (disks from I2A, Montpellier, France) on Mueller Hinton agar (Bio-Rad, Marne-la-Coquette, France) following the recommendations of the French committee for antimicrobial susceptibility testing based on the European Committee on Antimicrobial Susceptibility Testing (CASFM/EUCAST, May 2019) guidelines. Polymyxins: colistin MICs were measured by using the broth microdilution method (Biocentric, France). When Enterobacterales from a common sample shared the same antibiotic susceptibility pattern, they were assumed to be identical and only one was sent for sequencing.

## Molecular methods

Genomic DNA of 78 CRE isolates based on resistance pattern of antimicrobial susceptibility profile was extracted using the EZ1 DNA Tissue Kit (Qiagen, Courtaboeuf, France). Libraries

were prepared by using Nextera DNA Flex library preparation kit (Illumina, SanDiego, CA). The DNA was quantified using the Qubit Fluorometer (Thermo Fischer Scientific, Asnières sur Seine, France). Whole genome sequencing was performed on NextSeq 550 system (Illumina) with a mid-output kit (2x150 bases). Plasmids DNA of three different *E. coli* ST167 isolates were extracted with the Plasmid Midi Kit (Qiagen) for MinION system sequencing (Oxford Nanopore Technologies [ONT], Oxford, UK). Libraries were prepared with the Rapid Barcoding Kit, SQK-RBK004 (ONT) then plasmids DNA sequencing was performed on a Flongle Flow Cell (R9.4.1) on a MinION device (ONT).

## Multilocus sequence typing (MLST)

MLST profiles were checked by using MLST tool (v2.16.4) [26, 27]. Data were compiled and evaluated, and isolates were given allele numbers, and sequence types (STs). For *E. coli*, it was done using the Warwick MLST database (http://mlst.warwick.ac.uk/mlst/dbs/Ecoli) [28].

## Bioinformatic analysis

Reads quality was assessed by FastQC v0.11.8 [29]. Trim Galore v0.4.5 was used for quality and adapter trimming (reads with Phred quality score inferior to 20 and length inferior to 50 bases were discarded) [30]. MetaPhlAn2 was used to confirm the phenotypic identifications of the strains and to detect putative cross-contaminations [31]. Reads were assembled using SPAdes v3.11.1 [32]. The quality of the assemblies was examined by using QUAST v5.0.2 [33]. ARGs and plasmids replicons were searched by using Abricate v0.9.8 using AMRFinder database (2019-04-29) and PlasmidFinder database (2019-08-28), respectively, with 80% minimal coverage and nucleotidic identity filters [34, 35]. Chromosomal point mutations conferring quinolones resistance in *E. coli* and *K. pneumoniae* were searched with Resfinder 4.1 on the Center for Genomic Epidemiology website (http://www.genomicepidemiology.org)Core genomes were determined by multi-alignment tool Parsnp v1.2 [36]. The single nucleotide polymorphisms (SNPs) based phylogenetic tree was constructed with PhyML v3.0 using General Time Reversible (GTR) model and 1000 boostraps [37]. Plasmids were reconstructed by hybrid assembly of Nanopore and Illumina reads with Unicycler v0.4.9b [38]. ARGs and plasmids replicons were searched by Diamond v0.9.22.123 [39] with AMRFinder database (2019-04-29) and CD-HIT v4.7 [40] with PlasmidFinder database (2019-08-28), respectively. The Illumina reads of each isolate were mapped onto the reconstructed plasmid genomes with BWA v0.7.17 [41]. A plasmid was considered to be present in a genome if the Illumina reads covered >94% of the plasmid sequence with 10X depth and with maximal mapping quality. The 94% threshold was set according to the lowest coverage observed for the mapping of Illumina reads against the plasmid sequence within the same strain.

## Quantification of CRE

Rectal samples thoroughly mixed with 1 mL normal saline were serially diluted till $10^{-6}$ and plated onto MacConkey agar with 0.5 μg/mL ertapenem and without ertapenem and incubated overnight. Colony-forming units (CFU) were counted in decimal logarithms at the dilution in which 1 to 100 CFU were grown. The CRE relative abundance (CRE-RA) was calculated as the ratio of the CRE counts divided by the total number of Gram-negative bacteria, expressed as a percentage. For patients who were found to carry more than one CRE clone, the CRE-RA of the dominant clone was considered.

## Statistical analysis

Statistical analysis of relative abundance was performed using Wilcoxon and Kruskal-Wallis tests [42]. Differences of CRE-RA between community-acquired CRE and hospital-associated CRE were calculated by using Wilcoxon test, while Kruskal-Wallis test was used to calculate differences between community-acquired bacterial species and between the different community-acquired *E. coli* sequence types [43].

# Results

## Sampling

A total of 306 rectal swabs (one per patient) were collected within 24 hours of their admission to the traumatology and gynaecology wards of Benazir Bhutto hospital. Among them, 27.8% (n = 85) samples were from male patients from the traumatology ward, 21.9% (n = 67) from female patients from the traumatology ward and 50.3% (n = 154) were from female patients from the gynaecology ward. The average age for male and female trauma patients was 38 years and 44 years respectively, while for female gynaecology patients the average age was 28 years. From 94 patients who were subsequently hospitalised and in whom the first swab was negative on the ertapenem-supplemented MacConkey agar, a second swab was taken after 72 hours of their hospital stay (Fig 1). The mean duration between admission and follow-up sampling was eight days (range 3–60 days). Most of the swabs were collected during three periods of time: January to May 2018 (70/306, 23%), August to November 2018 (104/306, 34%), January to May 2019 (132/306, 43%) (Fig 2).

## Phenotypic characteristics of the recovered isolates

Enterobacterales isolates were Gram-negative, pink colored, short rods under the microscope. On MacConkey agar *E. coli* colonies appeared as pink, flat, dry and non-mucoid surrounded

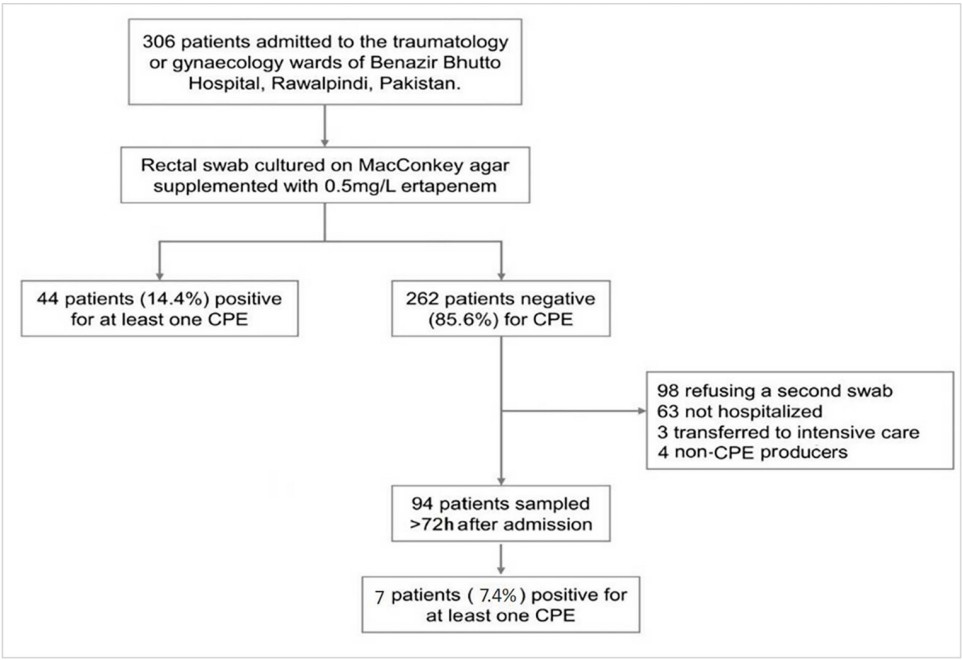

**Fig 1. Flowchart of the study.**

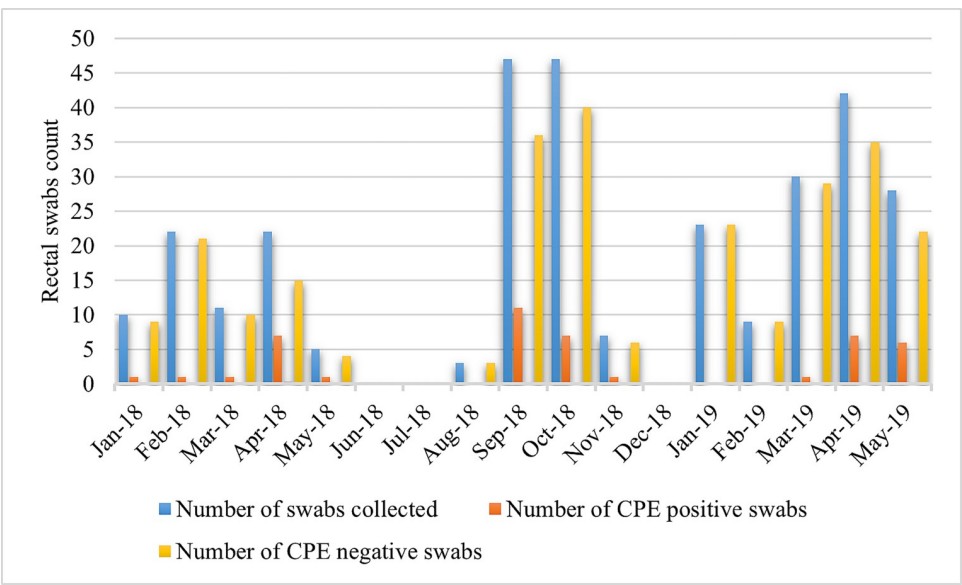

**Fig 2. Distribution of CPE positive and negative samples by month of collection, (January 2018 to May 2019).**

with darker pink zone of precipitated bile salts, *K. pneumoniae* were mucoid, and pink. *C. freundii*, and *E. cloacae* also formed pink colonies due to lactose fermentation. Biochemical tests results were interpreted according to Clinical Microbiology procedures handbook [24].

## Prevalence of CPE carriage

A total of 170 Enterobacterales isolates was cultured on ertapenem-supplemented MacConkey agar, among which 78 were left after deduplication (based on their antibiotic susceptibility profile). After genomic analysis, 73 were found to produce at least one carbapenemase-encoding gene. Among the 306 patients screened at admission, 48 were community acquired CRE, and 44 carried a CPE, yielding a CPE carriage rate of 14.4% (44/306) in the community (Fig 1). The carriage rate was 16.5% (14/85) among traumatology-admitted male patients, 19.4% (13/67) among traumatology-admitted female patients and 11.0% (17/154) among gynaecology-admitted female patients. All samples were from patients who were not showing any infectious disease symptoms and they were not receiving any medication at the time of sampling.

## Antimicrobial susceptibility testing

All the community-carried (n = 69) and hospital-acquired CPE isolates (n = 9) were found to be resistant to ceftazidime, cefotaxime, ertapenem, and cefoxitin while 1 strain (231A) remained apparently intermediate to cefepime. Imipenem was inactive against 92% strains while 79% isolates were resistant against aztreonam. For quinolones, 99% resistance was found against nalidixic acid and 97% against ofloxacin. Trimethoprim/sulfamethoxazole was found resistant against 94% strains, while a lower rate of 24% and 31% resistance was observed for amikacin and gentamicin, respectively. Only 1 isolate (201A) found resistant to colistin (S1 Table), MIC measured for this isolate was 4 mg/L and it was found phenotypically resistant to all antibiotics used except imipenem and amikacin. Bacterial isolates were grouped into extensively-drug resistant (XDR) and multi-drug resistant (MDR) based on the resistance phenotypes definitions of Magiorakos et al. [44]. According to these definitions, none of the isolates

was PDR, while 64 (82%) isolates were grouped into XDR and 14 (18%) isolates into MDR category (S2 Table).

## Antimicrobial resistance genes

A total of 78 CRE isolates were subjected to whole genome sequencing, among which 73 were carbapenemase producing Enterobacterales (CPE) while 5 strains did not carry any carbapenemase gene while phenotypically all isolates were resistant to carbapenems. Among the 73 Enterobacterales strains, 56 distinct antibiotic resistance genes were identified, including 17 encoding β-lactamases (Fig 3). The most common carbapenemase gene was $bla_{NDM-5}$ (n = 58) followed by $bla_{NDM-1}$ (n = 7), $bla_{NDM}$ (unidentified variants, n = 4), $bla_{OXA-181}$ (n = 3), $bla_{OXA-232}$ (n = 3) and $bla_{NDM-7}$ (n = 1). The three strains carrying $bla_{OXA-181}$ were *Citrobacter freundii* which also carried $bla_{NDM-1}$. The $bla_{NDM-5}$ gene was only found in *E. coli* while $bla_{OXA-232}$ was only found in *K. pneumoniae*. Among the hospital-associated strains (n = 9), eight carried $bla_{NDM-5}$ and one carried $bla_{NDM-1}$. Forty-three strains were found to harbour an ESBL-encoding gene, mainly CTX-M-15 ESBLs (one exception being CTX-M-139, which is a variant of CTX-M-15) (Fig 3 and S3 Table). The main plasmid-mediated quinolone resistance (PMQR) genes were *qnr*S1 (n = 12) and *qep*A (n = 3) (Fig 4). Consistent with their resistance phenotype, 66/68 *E. coli* had mutations in topoisomerases known to confer resistance to quinolones. On the other hand, no mutation was found in 2/68 quinolone-resistant *E. coli* and 3/3 quinolone-resistant *K. pneumoniae*. These five isolates carried PMQR genes (Fig 4). Multiple aminoglycoside resistance genes (ARG) coding for enzymes that inactivate the antibiotic were identified in 72/73 strains: *aadA2* (n = 46), *aadA1* (n = 19), *aadA5* (n = 12), *aph(3")-Ib* (n = 12), *aph(6)-Id* (n = 12), *aadA16* (n = 7), and *aac(3)-IIa* (n = 4) (Fig 5). Phenotypic resistance rates were low for amikacin and gentamicin since the antibiotic spectra of these ARGs are different. 16S rRNA methyltransferase genes, which confer resistance to all aminoglycosides, were found in 18 strains: *rmtB1* (n = 12), *rmtF1* (n = 3), *armA* (n = 3). The *sul* (dihydropteroate synthase) and *dfr* (dihydrofolate reductase) resistance genes that confer cotrimoxazole resistance were present in 67 and 73 strains, respectively, consistent with phenotypic resistance. Only one isolate (201A) carried the colistin resistance gene *mcr-1* (Fig 6). Same isolate was also found resistant to colistin *in vitro*. The correlation between the observed phenotypic antibiotic resistance pattern and genomic content of antibiotic resistance genes (ARGs) is presented in supporting information (S4 Table).

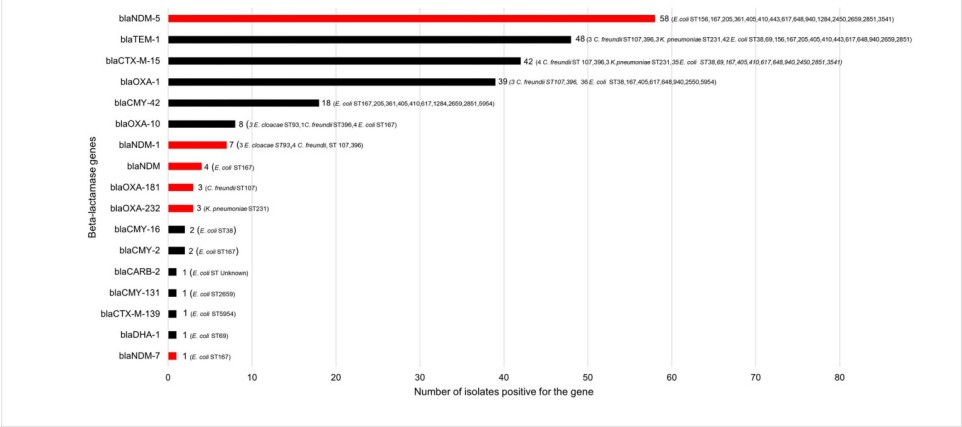

**Fig 3. Acquired beta-lactamase–encoding genes.** Carbapenemase-encoding genes are highlighted in red.

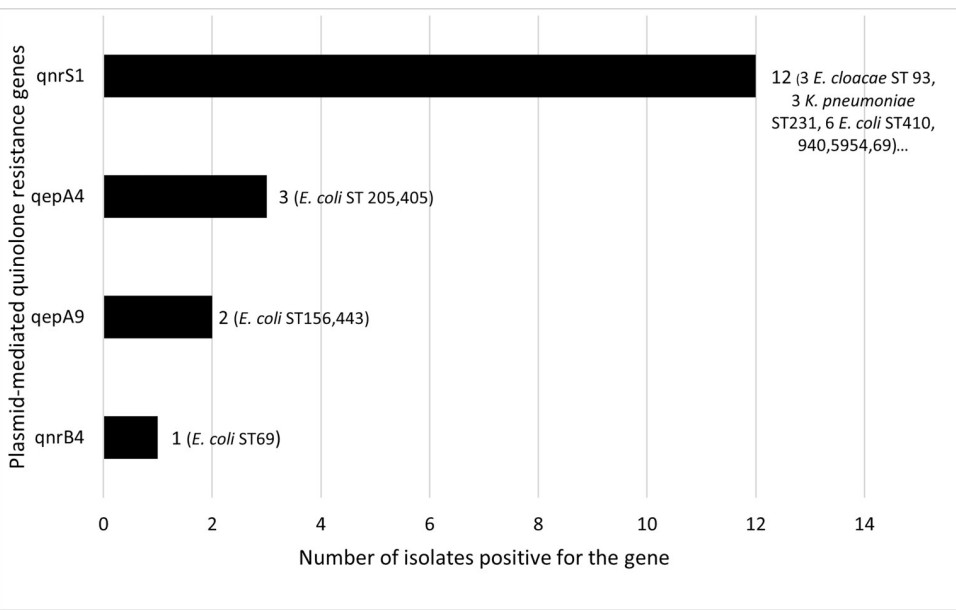

**Fig 4. Bar plot depicting the distribution of plasmid-mediated quinolone resistance genes.**

## Multilocus sequence typing (MLST)

MLST of community-acquired *E. coli* identified 18 different STs (S3 Table, Figs 6 and 7). The most prevalent were ST167 (phylogroup A, 14/60), ST405 (phylogroup D, 10/60), ST940 (phylogroup B1, 8/60), ST648 (phylogroup F, 6/60) and ST617 (phylogroup A, 5/60) (S3 Table). Hospital-associated *E. coli* (n = 8) were divided into five different STs except for two ST167

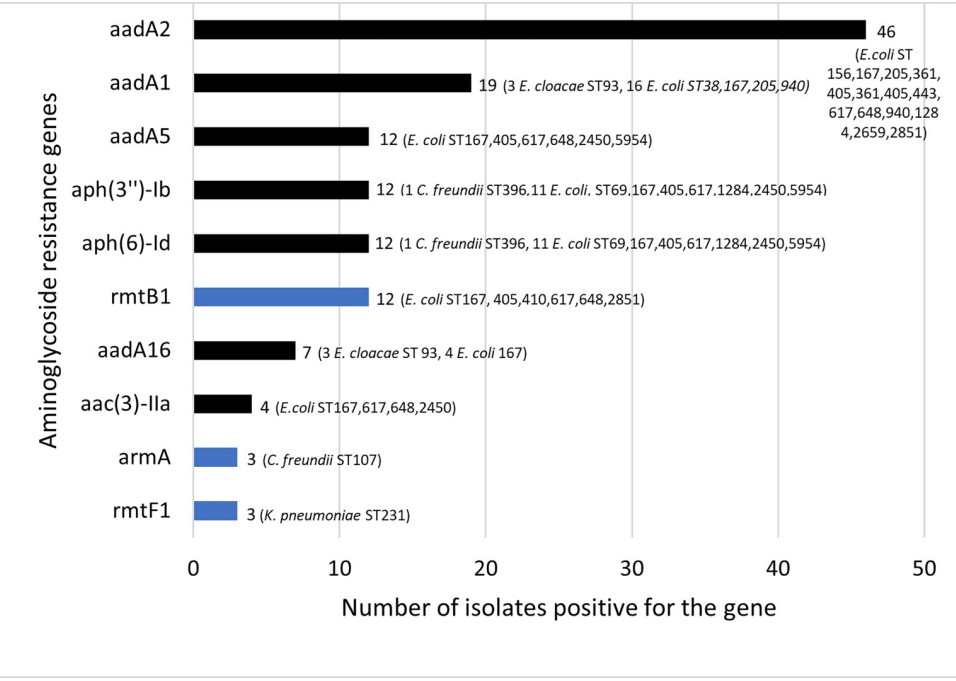

**Fig 5. Bar plot depicting the distribution of aminoglycoside resistance–encoding genes.**

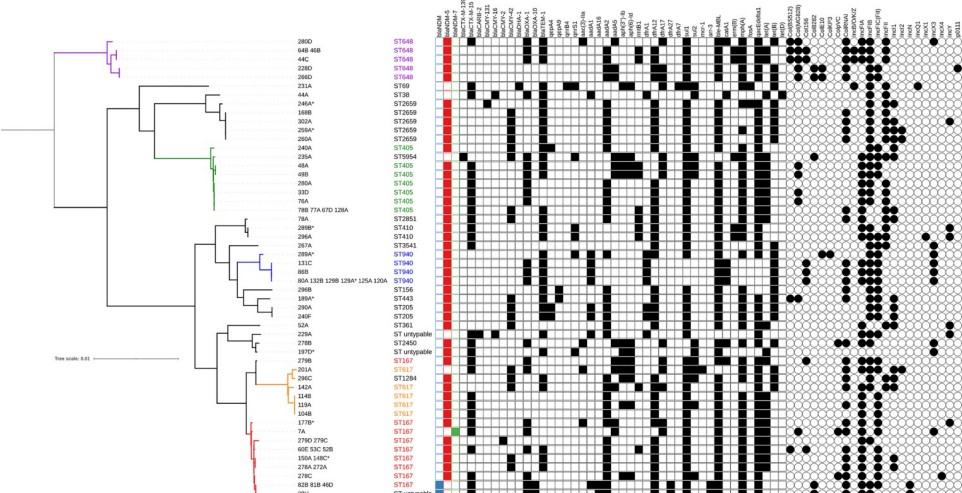

**Fig 6. Phylogeny, STs, resistance genes and plasmids of *E. coli*.** *: hospital-acquired isolates.

strains and two ST2659 strains (Fig 6). The three *K. pneumoniae* in the study were isolated from three different community subjects and all belonged to ST231. The three *Enterobacter cloacae* in the study (two community-acquired, one hospital-associated) belonged to ST93. The four *C. freundii* were of community origin and belonged to ST396 (n = 1) and ST107 (n = 3).

## Phylogeny

The core genome of the 68 *E. coli* was made of 2,094 genes (1,807,969 bases in total). The five most frequent STs of community-acquired *E. coli* made distinct clades (Fig 8). The maximal genetic distances between isolates of the same clade were 1,954 SNPs for ST167, 1,606 SNPs for ST405, 3,098 SNPS for ST940, 3,101 SNPs for ST617 and 3,679 SNPs for ST648. Among all the *E. coli*, the most genetically distant were separated by 37,852 SNPs. Phylogenetic analysis revealed that community and hospital acquired were closely related isolates suggesting that the community reservoir of CPE is fueling the hospital reservoir. Among ST940 and ST167 community and hospital acquired isolates were clonal sharing less than 10 SNPs.

## Replicon typing and plasmid analysis

We identified 27 plasmid replicons in the 78 genomes. The IncX3 and IncFII incompatibility groups commonly associated with $bla_{NDM}$ were present in 23% (18/78) and 41% (32/78) strains, respectively (Fig 9). To assess the diversity of plasmids within the ST167 strains, we sequenced and reconstructed plasmids from three different ST167 *E. coli* isolates (46D, 60E, and 279D) that had different genetic backgrounds based on the resistance genes, phylogenetics and plasmids typing results. They were found to carry $bla_{NDM}$ on three different IncF plasmids and isolate 46D possessed $bla_{NDM}$-encoding IncF plasmid with another $bla_{NDM}$-encoding IncN replicon plasmid. Then, we mapped the Illumina reads of all the *E. coli* ST167 on the reconstructed plasmids sequences. The four reconstructed plasmids were also found in the genomes of the other *E. coli* ST167 (Fig 10).

## Relative abundance

The mean relative abundance for CRE was 0.5% (median 0.1%, min 0.0003% and max 3.4%). There were no significant differences of CRE-RA between community-acquired CRE and

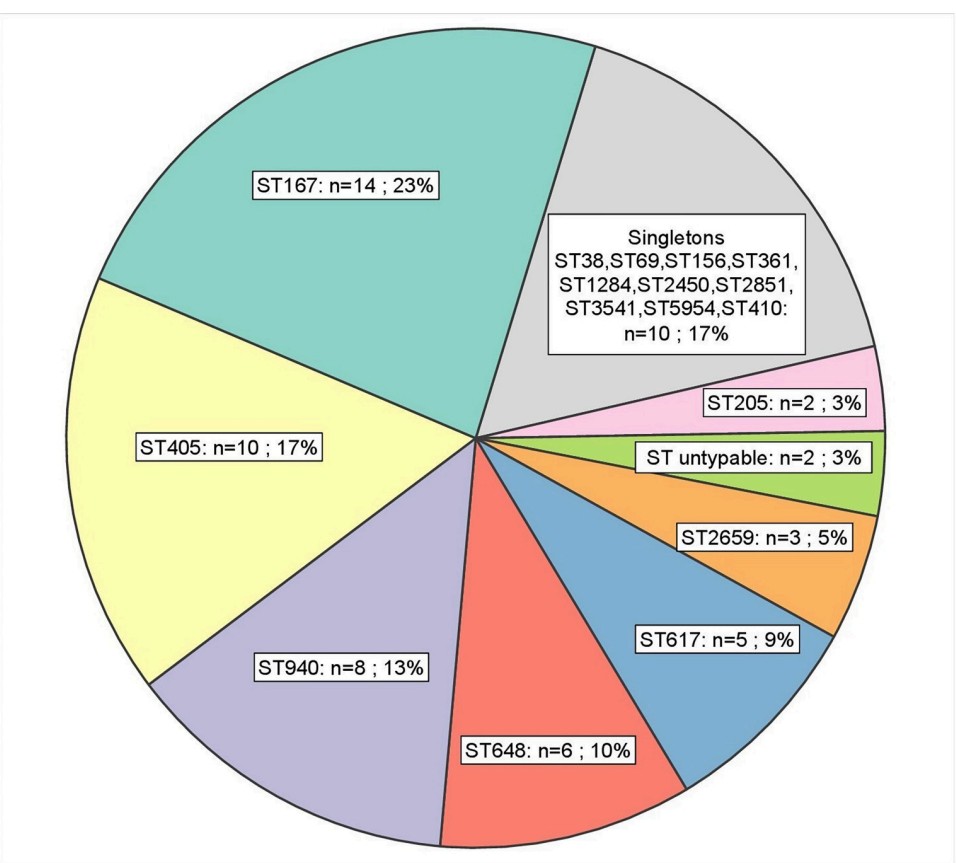

**Fig 7. Pie chart of the sequence types (ST).** The distribution of STs among carbapenemase-producing *E. coli* from the community.

hospital-associated CRE (Wilcoxon test, p = 0.2) (Fig 11), between community-acquired bacterial species (Kruskal-Wallis test, p = 0.29) (Fig 12) and between the different community-acquired *E. coli* sequence types (Kruskal-Wallis test, p = 0.92) (Fig 13).

## Discussion

The main result of this study is the high prevalence of CPE carriage (14.4%, 44/306) among patients attending the traumatology and gynaecology wards of the Benazir Bhutto hospital of Rawalpindi. This high prevalence rate is in line with previous reports [20, 21, 45]. High community carriage can lead towards longer hospitalization, higher medical cost, and increased mortality due to limited treatment options. Our study results suggest that community carriage is also one of the important factors for CPE dissemination in hospital settings. A second sample was taken from 94 patients (negative at admission) during their hospital stay and it was found that 7 (7.4%) acquired a CPE. Phylogenetic analysis revealed that there is genetic relatedness of hospital acquired isolates with few community acquired isolates suggesting a person to person spread or same source of contamination. Same clones were circulating in community and hospital settings. In hospital environment, antimicrobial exposure, longer hospitalization, the use of invasive devices, and severe underlying infections are considered independent risk factors for carbapenem producing Enterobacterales, but our study shows increasing CPE carriage among community along with the detailed genomic analysis to understand the determinants responsible for the spread of CPE specially in community settings. Different studies

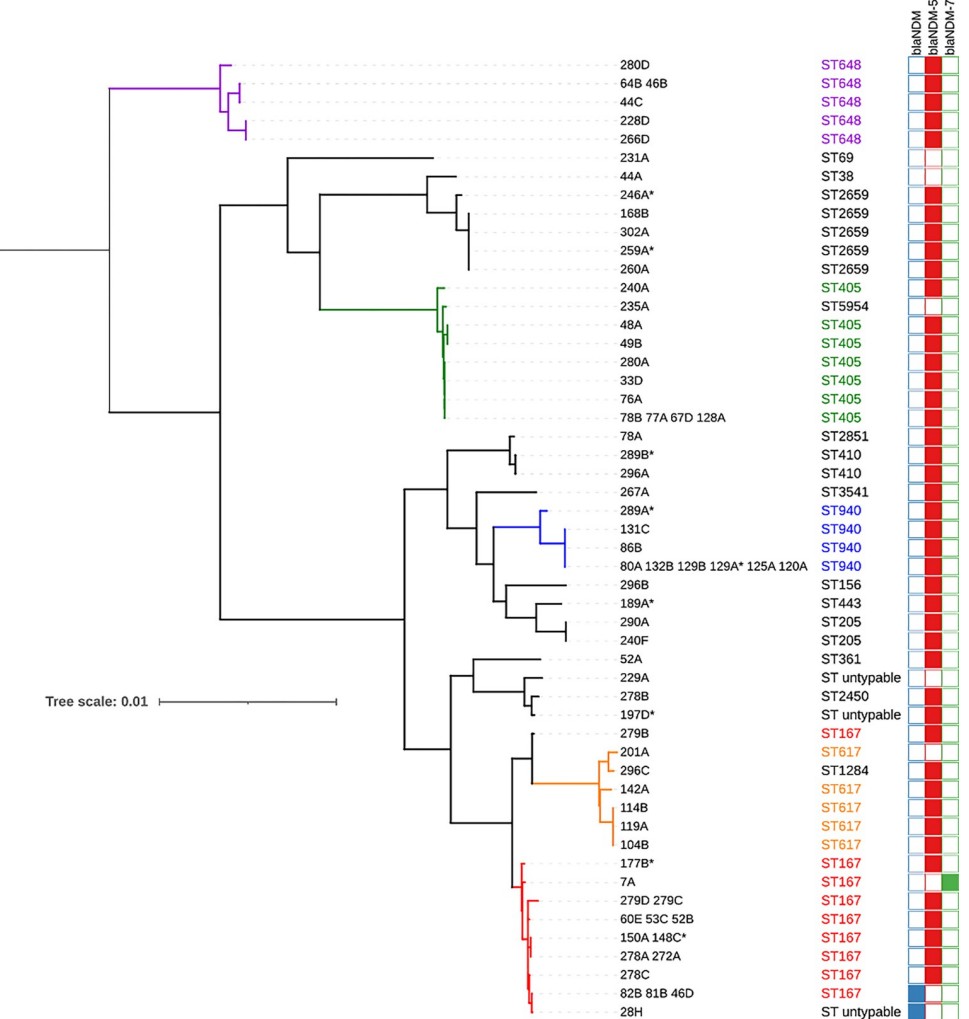

**Fig 8. Phylogenetic tree of *E. coli*.** *: hospital-associated isolates. The dendrogram showing the phylogenetic relationship between community-acquired and hospital-associated *E. coli*.

including a review of 10 studies of different regions (Asia, Europe, and North America) reported prevalence of CRE ranging from 0.4% to 29.5% isolated from various samples including faecal samples from healthy individuals and outpatients showing that community-based CPE are increasing in Asia, particularly in Indian subcontinent [46–48]. In 2013, Nair et al, from India reported 12.3% prevalence of carbapenem resistance among Enterobacterale isolates from patients admitted to different wards, ICU, and visiting outpatient department in respective hospital [49]. Two main factors underlying the specific situation of CPE in Pakistan may be proposed. The first one is the uncontrolled use of antibiotics, several antibiotics can be purchased over-the-counter without prescription, thereby promoting self-medication and potential overuse, causing a big challenge to community health [50]. The second factor is that Pakistan is a developing country where access to clean drinking water and safe food is not secured for large part of population, which may facilitate the circulation of intestinal bacteria such as CPE. A study conducted in Pakistan on environmental water collected from different sites reported the presence of NDM-1 producing bacteria in water sources [51]. In India, the $bla_{NDM}$ was similarly found in tap water [52].

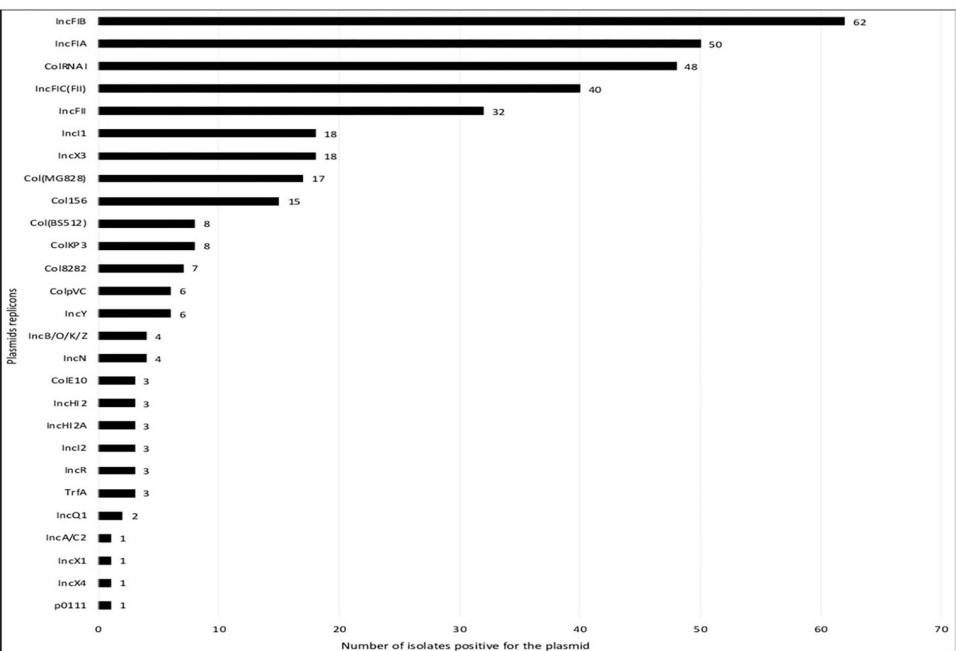

**Fig 9. Distribution of plasmids replicons.**

The diffusion of NDM in the community of Rawalpindi was driven by a limited number of *E. coli* STs, with 43/55 (78%) of NDM-producing *E. coli* belonging to the ST167, ST405, ST940, ST648, and ST617. The most frequent ST was ST167 which belongs to the phylogroup A. ST167 NDM-producing *E. coli* had already been spotted [53–55], yet the extent of its diffusion in this region had not been assessed to date. An Indian study conducted in 2020, on sequence

| | Species | ST | 60E *E. coli* ST167 IncF | 279D *E. coli* ST167 IncF | 46D *E. coli* ST167 IncN | 46D *E. coli* ST167 IncF |
|---|---|---|---|---|---|---|
| 279B | *Escherichia coli* | 167 | | | Present | Present |
| 46D | *Escherichia coli* | 167 | | | Present | Present |
| 81B | *Escherichia coli* | 167 | | | Present | Present |
| 82B | *Escherichia coli* | 167 | | | Present | Present |
| 177B | *Escherichia coli* | 167 | Present | | | |
| 52B | *Escherichia coli* | 167 | Present | | | |
| 53C | *Escherichia coli* | 167 | Present | | | |
| 60E | *Escherichia coli* | 167 | Present | | | |
| 148C | *Escherichia coli* | 167 | | | | |
| 150A | *Escherichia coli* | 167 | | | | |
| 272A | *Escherichia coli* | 167 | | | | |
| 278A | *Escherichia coli* | 167 | | | | Present |
| 278C | *Escherichia coli* | 167 | | | | |
| 279C | *Escherichia coli* | 167 | | Present | | |
| 279D | *Escherichia coli* | 167 | | Present | | |
| 7A | *Escherichia coli* | 167 | | | | |

**Fig 10. Screening of the four reconstructed plasmids within *E. coli* ST167 isolates.**

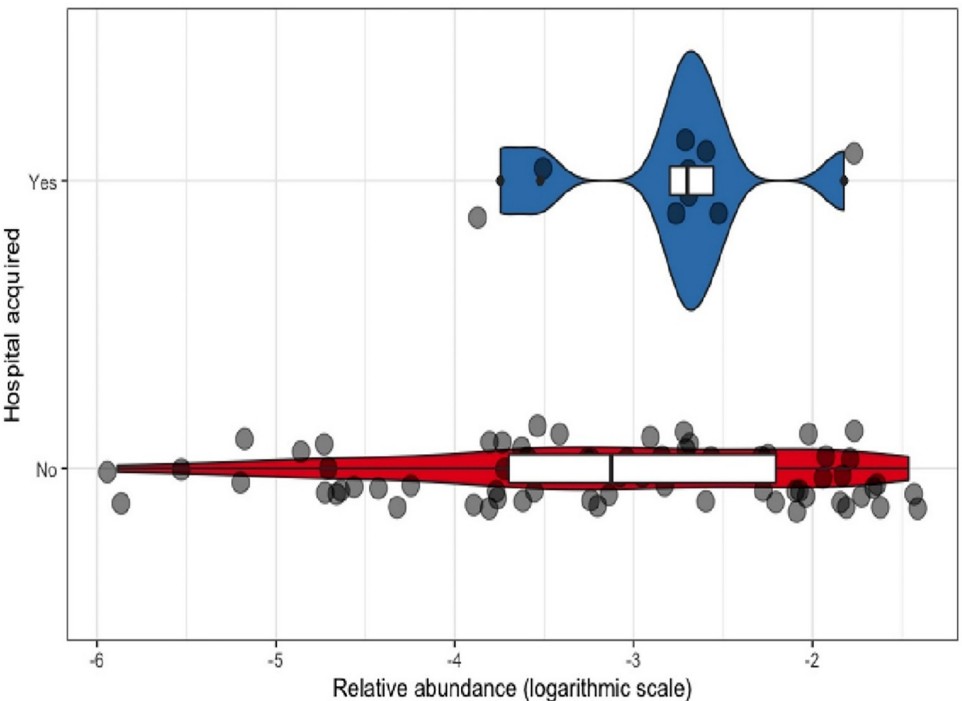

**Fig 11. CRE relative abundance of community-acquired and hospital-acquired CRE.**

types of cefotaxime resistant *E. coli* showing that ST167 was more prevalent in the community followed by ST410 and ST648 [56]. Another study conducted in sewage from Islamabad showed that ST648 was among prevalent STs in *E. coli* isolates [22]. A Chinese study analysed all bacterial genomes in the GenBank database containing *bla*$_{NDM}$ [13]. Among these

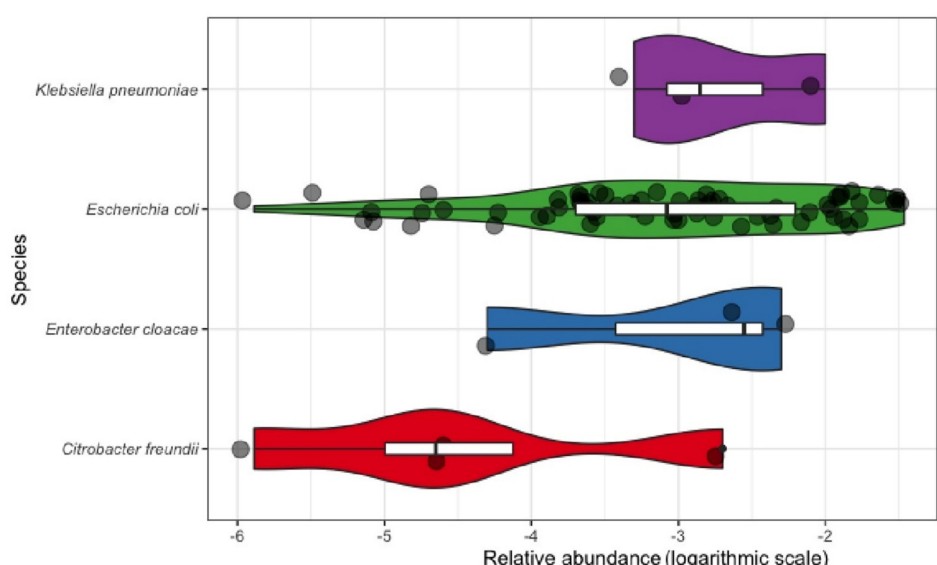

**Fig 12. CRE relative abundance of the different species of *Enterobacterales*.**

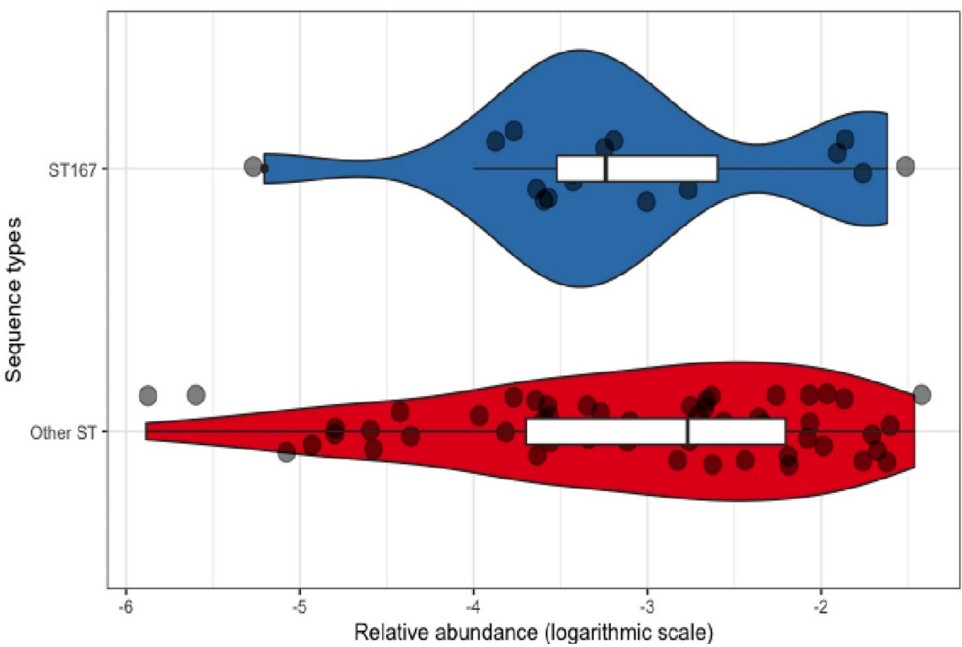

**Fig 13. CRE relative abundance of *E. coli* ST167 versus other ST of *E. coli*.**

genomes, *E. coli* was the most common species (n = 305) and the ST167 was the main ST (44/305) among those possessing $bla_{NDM}$. In addition, *E. coli* strains of ST167 carrying $bla_{NDM}$ have been reported from China (hospital-outbreak), India, South Korea, Switzerland, South Africa, and the United States [13, 57]. ST167 carrying $bla_{NDM-5}$ has been the most prevalent sequence type reported from China in clinical settings and has potential to become the main ST type worldwide [58, 59].

Combination of resistance genes, and ability to easily disseminate put *E. coli* ST167 as a potential "high-risk clone" [60] such as the CTX-M-15-producing *E. coli* ST131. Moreover, using nanopore sequencing technology, we were able to retrieve circularised plasmids for *E. coli* ST167 strains and among these isolates, we observed several $bla_{NDM}$-encoding plasmids. The diversity of resistance plasmids observed within ST167 isolates indicates that carbapenem resistance is not spreading through a specific plasmid and that such as *E. coli* ST131, there does not seem to be a bottleneck for the $bla_{NDM}$–bearing plasmids for the *E. coli* ST167.

The genomic analysis showed that $bla_{NDM}$ (90%) was the most prevalent carbapenemase gene in our study and $bla_{NDM-5}$ was the most common variant (58/70, 82%). Previously, a study from Lahore (Pakistan) showed the presence of $bla_{NDM-5}$ in isolates from neonates [61]. However, majority of studies showed that $bla_{NDM-1}$ was more prevalent in Asia especially Indian subcontinent [13, 51, 57]. $Bla_{NDM-5}$ seems to be becoming a prevalent NDM variant in the Indian subcontinent [14]. Apart from NDMs, the OXA-181 carbapenemase was found in three community acquired *C. freundii* belonging to ST107 from three different patients. These three strains were co-harbouring $bla_{NDM-1}$ and $bla_{OXA-181}$ and their resistance genes and plasmid content were also similar. Previous studies in Pakistan have reported the presence of the *mcr*-1 gene in migratory birds, clinical settings, and healthy broilers [62–64]. We report here the first detection of *mcr-1* from an outpatient in Pakistan. In countries like Pakistan, where the prevalence of CPE is high, increasing resistance against last-line antibiotics such as colistin is of major concern. Spread of *mcr* genes in CRE is a potential concern as it could create CRE strains that would be potentially pan-drug resistant. Although only one CRE isolate in this

study was found to be carrying colistin resistance gene, it signifies that surveillance and monitoring should be conducted on regular basis to slow down the emergence of such strains. A study from Thailand also reported similar findings of *mcr-1* detection from clinically isolated CRE during 2016–2019 [65]. Dissemination of *mcr-1* together with $bla_{NDM}$ in Asian counties is alarming and it needs attention.

The average age for male and female trauma patients was 38 years and 44 years respectively, while for female gynaecology patients the average age was 28 years. According to a study age >50 years is one of the risk factors for carbapenem resistance [66]. Another study has reported high association of carbapenem resistance acquisition in children with age less than ten years [67]. Hence, we assessed the prevalence of CPE in a group with no specific risk factor for multidrug resistance carriage. There is possibility that the CPE prevalence could even be higher in specific populations such as young children or elderly people.

The mean relative abundance of CRE (CRE-RA) was similar to that found in other studies in Europe for community ESBL-producing *E. coli* [68, 69] highlighting that in patients with no recent exposure to antibiotics, multidrug-resistant Enterobacterales remain subdominant. Surprisingly though, the mean CRE-RA of hospital-acquired CRE was not higher. Still, we did not record the antibiotic intake after admission in these patients.

This study will be helpful in raising awareness among medical professionals, scientific community, and policy makers about recent antimicrobial resistance trend in community and need for solutions by limiting the use of antimicrobials and switching towards alternative options such as, vaccines, immunotherapeutics, and modulation of gut microbiota [70].

This work has certain limitations. The population considered might not be a true reflection of community, even though we only included patients who reported no antibiotic intake nor hospitalisation in the 3-months preceding the visit. Also, we used rectal swabs instead of faeces for feasibility matters which may have resulted in underestimation of CPE carriage. Accordingly, we may have overestimated the hospital acquisition rate in that some patients deemed negative for CPE at admission may have been false negative. Screening on ertapenem supplemented MacConkey agar may have led to underestimation of total Enterobacterales as a substantial number of phenotypically susceptible (may carry genotypic resistance markers) strains may have been excluded.

## Conclusion

We observed a high carriage rate of CPE among community patients in Rawalpindi, mostly driven by a limited number of *E. coli* STs among where ST167 was dominant. Asymptomatic CPE carriers are the potential source of its dissemination in community, hospitals as well as in environment as phylogenetic analysis indicated that same CPE clones were circulating in community and hospital settings. Successful dissemination of Enterobacterales clones in community, continuous transmission of carbapenemase harboring plasmids through mechanism of horizontal gene transfer, and co-occurrence of multi-drug resistance genes, are the main reasons for increased carbapenem resistance in community. Stopping dissemination is important otherwise, CPE will achieve a foothold in community and hospitals. CPE transmission can be controlled by active surveillance, raising awareness, regular screening, and implementation of infection control strategies.

## Supporting information

**S1 Table. Antimicrobial susceptibility testing results.**
(DOCX)

**S2 Table. Categorization of CRE isolates into XDR and MDR.**
(DOCX)

**S3 Table. Acquired antibiotic resistance genes, sequence types, species, and phylogroups.** *:
hospital-acquired isolates.
(XLSX)

**S4 Table. Correlation between the genomic content of antibiotic resistance genes (ARG)
and observed phenotypic resistance.**
(DOCX)

## Acknowledgments

We thank the patients, medical and paramedical staff of Benazir Bhutto hospital, Rawalpindi,
Pakistan for cooperation during sampling.

## Author Contributions

**Conceptualization:** Etienne Ruppé, Rabaab Zahra.

**Data curation:** Stéphane Lo, Shaheen Akhtar Malik.

**Formal analysis:** Amina Habib, Stéphane Lo, Khanh Villageois-Tran, Marie Petitjean, Etienne
Ruppé.

**Funding acquisition:** Etienne Ruppé, Rabaab Zahra.

**Investigation:** Amina Habib, Shaheen Akhtar Malik, Etienne Ruppé, Rabaab Zahra.

**Methodology:** Amina Habib, Stéphane Lo, Marie Petitjean.

**Project administration:** Etienne Ruppé, Rabaab Zahra.

**Resources:** Shaheen Akhtar Malik, Laurence Armand-Lefèvre, Etienne Ruppé, Rabaab Zahra.

**Software:** Khanh Villageois-Tran, Marie Petitjean.

**Supervision:** Laurence Armand-Lefèvre, Etienne Ruppé, Rabaab Zahra.

**Validation:** Stéphane Lo, Marie Petitjean.

**Visualization:** Amina Habib, Marie Petitjean.

**Writing – original draft:** Amina Habib, Stéphane Lo, Etienne Ruppé, Rabaab Zahra.

**Writing – review & editing:** Amina Habib, Stéphane Lo, Khanh Villageois-Tran, Marie Petit-
jean, Laurence Armand-Lefèvre, Etienne Ruppé, Rabaab Zahra.

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
