## [Decision Letter · Decision Letter 0]

6 Dec 2021

PONE-D-21-36109Dissemination of Carbapenemase-producing Enterobacterales in the community of Rawalpindi, PakistanPLOS ONE

Dear Dr. Zahra,

Thank you for submitting your manuscript to PLOS ONE. After careful consideration, we feel that it has merit but does not fully meet PLOS ONE’s publication criteria as it currently stands. Therefore, we invite you to submit a revised version of the manuscript that addresses the points raised during the review process.

We look forward to receiving your revised manuscript.

Kind regards,

Zhi Ruan, Ph.D.

Academic Editor

PLOS ONE

Journal Requirements:

" ext-link-type="uri" xlink:type="simple">https://journals.plos.org/plosone/s/file?id=ba62/PLOSOne_formatting_sample_title_authors_affiliations.pdf"

This work was partially supported by the “Fondation pour la Recherche Médicale” (Equipe FRM 2016, grant 325 number DEQ20161136698); the Direction Générale des Armées (project FastGeneII); the Programme Hubert Curien “Peridot” (Ministère des Affaires Etrangères, France) and by the Higher Education Commission, Pakistan under Pak-France Joint Grant (PERIDOT) to RZ and ER and International Research Support Initiative Program (IRSIP) (PIN: 43 BMS 16) to AH. 

This work was partially supported by the “Fondation pour la Recherche Médicale” (Equipe FRM 2016, grant 325 number DEQ20161136698); the Direction Générale des Armées (project FastGeneII); the Programme Hubert Curien “Peridot” (Ministère des Affaires Etrangères, France) and by the Higher Education Commission, Pakistan under Pak-France Joint Grant (PERIDOT) to RZ and ER and International Research Support Initiative Program (IRSIP) (PIN: 43 BMS 16) to AH. 

Reviewers' comments:

Reviewer's Responses to Questions

**Comments to the Author**

1. Is the manuscript technically sound, and do the data support the conclusions?

Reviewer #1: Yes

Reviewer #2: Partly

Reviewer #3: Yes

2. Has the statistical analysis been performed appropriately and rigorously? 

Reviewer #1: Yes

Reviewer #2: I Don't Know

Reviewer #3: N/A

3. Have the authors made all data underlying the findings in their manuscript fully available?

Reviewer #1: No

Reviewer #2: Yes

Reviewer #3: Yes

4. Is the manuscript presented in an intelligible fashion and written in standard English?

Reviewer #1: Yes

Reviewer #2: Yes

Reviewer #3: Yes

5. Review Comments to the Author

Reviewer #1: I have some question/comments for the author to improve their manuscript.

1. Introduction: Situation of CPE/CRE in Pakistan should be mentioned.

2. Line 110-118: How many isolates were selected? and what is the criteria to select?

3. Line 178-185: What is colistin susceptibility result? it is not mention in this part.

4. Line 230: % should be shown.

5. Discussion: Although this study showed only one case/isolate of mcr-harboring CPE but it is a signal to monitoring. The mcr-harboring CRE were documented in several literature. The author can discuss this detail to lead awareness of the clinician and infection control. Please see 'Paveenkittiporn et al., Whole-Genome Sequencing of Clinically Isolated Carbapenem-Resistant Enterobacterales Harboring mcr Genes in Thailand, 2016-2019. Front Microbiol. 2021 Jan 11;11:586368. doi: 10.3389/fmicb.2020.586368.". This might be useful to discuss because it report from the Asian country.

Reviewer #2: Reporting data on the detection of CPE in each country is necessary to understand the dissemination of CPE and to develop countermeasures. However, the description of the results is not clear in the text. It makes difficult to understand results and needs to be revised.

L. 186 - 208

The number of CPE is unclear. How many CPE strains were isolated, 73, or 78? Which genes were detected in the strains? Which species were detected? Authors should add a table summarizing the essential information from Fig. S3 to the text.

Methods, Patients and samples

Although the paper is about CPE in the Community, it includes strains detected after 72 hours of hospitalization. These should be excluded.

L. 142 - 150

There is a problem with CRE quantification because the counts are made using cultured bacteria that have been previously subjected to ertapenem selection, which overestimates the number of CRE. Also, CRE quantification is not used for discussion and I do not understand the significance of its inclusion. It would be better to delete the descriptions and figures related to CRE quantification.

L. 215

Strains named Enterobacter cloacae complex should be re-identified using ANI and the correct species name filled in. Also, do not abbreviate the name of a genus that appears for the first time.

Fig.1

In the process of processing the 262 negative patients, 4 patients (negative:262-refused:98-not-hospitalized:63-ICU:3-sampled:94) are missing. Please check.

Fig.3

It should be carefully explained where the "clade" is. Especially for the ST648 clade, it is difficult to understand where the clade is located on the tree. It would be better to add some outgroups and make root for easier reading.

In the stacked bar graph of Fig. S1, the number of samples and the results are stacked on top of each other, and the number may be twice as large. It needs to be redesigned.

In Fig. S2, S4, and S5, only the number of detections is listed, and information such as from which ST type of which bacteria species is not available. Please add this information.

Reviewer #3: In this study the authors investigated the prevalence of carbapenemase-producing Enterobacterales (CPE) in patients attending a hospital from the community in Rawalpindi, Pakistan. Rectal swabs obtained from patients attending traumatology and gynaecology services without any preceding antibiotic exposure, hospitalisation or travel history in the last 3 months were plated on screening agar. Enterobacterales were identified by MALDI-TOF MS and antibiotic susceptibility testing and whole-genome sequencing performed.

The paper is well written and all the results are clearly presented. It presents valuable information on the prevalence of community CPE in Pakistan. WGS highlights the dominant sequence types carrying CPE genes and provides epidemiological information on plasmids carrying carbapenemase genes.

I have the following comments and suggestions for the authors:

METHODS:

Page 5, line 98: Antimicrobial susceptibility testing. It would add to the manuscript if the authors could perform susceptibility testing against newer agents, like aztreonam+avibactam and cefiderocol.

RESULTS:

Sampling: Page 8, line 160-162: In this study a young patient cohort was investigated (average age 38 and 44 years). How does this compare to other studies from the region? Could this influence the finding?

Prevalence of CPE carriage: Page 8, line 169: It would be interesting to add some information on whether the high community prevalence was reflected in disease burden in the hospital. How many of the patients required treatment for CPE? Were any of the hospital acquired cases considered to be linked?

Phylogeny: Page 10, line 220: The authors comment on the maximal genetic distances but do not discuss genetic similarities. On the tree some of the sequences look like they are related? Did they find possible relatedness between community and hospital acquired strains indicating possible cross transmission? Could the authors comment on this?

Relative abundance: Page 11, line 239: The authors present data on quantification of CRE but do not discuss the finding. Did they expect a difference in the relative abundance between community-acquired CRE and hospital-associated CRE? Does this depend on whether the patient receives antibiotics? I suggest to briefly mention this in the discussion.

DISCUSSION:

Line 248: ‘CPE carriage is mostly associated with hospital environment’ Did the other studies undertaken in Pakistan focus on hospital acquired cases? It would be interesting to discuss isolates that were acquired in hospital here. On the tree they look spread out over the clades but closely related to some of the community isolates? What is known about hospital acquisition of CPE in hospitals in Pakistan? How many of the CPE positive patients did require CPE treatment? What impact does the high community carriage rate have on hospitals?

Line 278: ‘Combination of resistance genes, virulence, and ability to easily disseminate…’ The authors mention virulence here but do not discuss virulence data for the observed ST types. How frequent is ST 167 found in clinical samples?

Line 295/296: It would add to the value of the manuscript to be able to discuss cefiderocol susceptibilities here. It might not be available as an agent in Pakistan yet but with the high prevalence of metallo-b-lactamases it would be helpful to assess its activity against the isolates of the study.

6. PLOS authors have the option to publish the peer review history of their article (what does this mean?). If published, this will include your full peer review and any attached files.

Reviewer #1: No

Reviewer #2: No

Reviewer #3: **Yes: **Kirsten Schaffer

---

## [Author Response · Author response to Decision Letter 0]

21 Jan 2022

RE: PLOS One 

Title: Dissemination of Carbapenemase-producing Enterobacterales in the community of Rawalpindi, Pakistan

Reviewer's comments:

Reviewer #1: I have some question/comments for the author to improve their manuscript.

1. Introduction: Situation of CPE/CRE in Pakistan should be mentioned.

Response: In revised version of manuscript, the information regarding CPE situation in Pakistan has been added. Please see lines 73 – 75 at page No. 4.

2. Line 110-118: How many isolates were selected? and what is the criteria to select?

Response: No. of isolates and selection criteria have been mentioned at page No. 5 (lines 113 -114).

3. Line 178-185: What is colistin susceptibility result? it is not mention in this part.

Response: Colistin susceptibility results have been added on Page No. 9 (lines189-191). 

4. Line 230: % should be shown.

Response: The percentages of IncX3 and IncFII incompatibility groups commonly associated with blaNDM are mentioned in revised version on Page 11 (line 242).

5. Discussion: Although this study showed only one case/isolate of mcr-harboring CPE but it is a signal to monitoring. The mcr-harboring CRE were documented in several literature. The author can discuss this detail to lead awareness of the clinician and infection control. Please see 'Paveenkittiporn et al., Whole-Genome Sequencing of Clinically Isolated Carbapenem-Resistant Enterobacterales Harboring mcr Genes in Thailand, 2016-2019. Front Microbiol. 2021 Jan 11;11:586368. doi: 10.3389/fmicb.2020.586368.". This might be useful to discuss because it report from the Asian country.

Response: We have added details as suggested and thanks to the reviewer for suggesting this article. In the revised version, we have cited this paper in the discussion, to highlight the spread of mcr harboring CRE in Asian countries. (Page 15, lines 319-325).

Reviewer #2:

Reporting data on the detection of CPE in each country is necessary to understand the dissemination of CPE and to develop countermeasures. However, the description of the results is not clear in the text. It makes difficult to understand results and needs to be revised.

Response: Your comments have been considered and revisions have been made throughout the text to have a clearer reading of the results.

L. 186 - 208

The number of CPE is unclear. How many CPE strains were isolated, 73, or 78? Which genes were detected in the strains? Which species were detected? Authors should add a table summarizing the essential information from Fig. S3 to the text.

Response: We have clearly mentioned the number of CPE in revised version. Page No. 9, lines 193-195. Fig. S3 has been converted to table (S1 Table) containing all essential information such as genes detected, species, sequence types and phylogroups. 

Methods, Patients and samples

Although the paper is about CPE in the Community, it includes strains detected after 72 hours of hospitalization. These should be excluded.

Response: Yes, the paper mainly discusses the community carriage of CPE but we also followed up (where possible) for the acquisition of CPE during hospitalization. We consider that this data is also of value as it shows the genetic features of hospital associated isolates which were quite like community acquired isolates. Also, the reviewer # 3 has suggested to include the discussion of hospital acquired isolates, hence we are keeping this information in the revised version. 

L. 142 - 150

There is a problem with CRE quantification because the counts are made using cultured bacteria that have been previously subjected to ertapenem selection, which overestimates the number of CRE. 

Response: The samples have been cultured on ertapenem-supplemented media and have not been exposed to ertapenem prior to spreading onto the plate. 

Also, CRE quantification is not used for discussion and I do not understand the significance of its inclusion. It would be better to delete the descriptions and figures related to CRE quantification.

CRE quantification has been included in Discussion. Page 15-16, L:333-337

L. 215

Strains named Enterobacter cloacae complex should be re-identified using ANI and the correct species name filled in. Also, do not abbreviate the name of a genus that appears for the first time.

Response: ANI of the three Enterobacter strains was calculated with FastAni (Jain, C., Rodriguez-R, L.M., Phillippy, A.M., Konstantinidis, K.T. Aluru, S. Nat Commun 9, 5114 (2018), and following were the results:

ANI of 99.9369 for 240Ea with Enterobacter cloacae,

ANI of 99.9451 for 255A with Enterobacter cloacae

ANI of 99.934 for 259B with Enterobacter cloacae. 

Hence the three strains are E. cloacae and changes have been made accordingly in the revised manuscript. (Page No. 3, line 71), (Page No. 9, line 199) and (Page No. 11, line 224). 

Fig.1

In the process of processing the 262 negative patients, 4 patients (negative:262-refused:98-not-hospitalized:63-ICU:3-sampled:94) are missing. Please check.

Response: These 4 patients were found carbapenem resistant on ertapenem supplemented agar but were not carrying any carbapenemase gene (CPE negative 262: refused:98-not hospitalized: 63-ICU:3-sampled:94,while remaining 4 were CRE not CPE, hence missing). We have made changes in Fig. 1 to clear this confusion. 

Fig.3

It should be carefully explained where the "clade" is. Especially for the ST648 clade, it is difficult to understand where the clade is located on the tree. It would be better to add some outgroups and make root for easier reading.

Response: Changes have been made as suggested in Fig. 3.

In the stacked bar graph of Fig. S1, the number of samples and the results are stacked on top of each other, and the number may be twice as large. It needs to be redesigned.

Response: Fig. S1 has been redesigned. Number of samples and results have been separated.

In Fig. S2, S4, and S5, only the number of detections is listed, and information such as from which ST type of which bacteria species is not available. Please add this information.

Response: All the information of the isolates (ST types, bacterial species) has been provided in detail in Table S1,while the ST types and species have also been updated in Figures S2, S3 and S4 as per reviewer’s suggestion.

Reviewer #3: 

In this study the authors investigated the prevalence of carbapenemase-producing Enterobacterales (CPE) in patients attending a hospital from the community in Rawalpindi, Pakistan. Rectal swabs obtained from patients attending traumatology and gynaecology services without any preceding antibiotic exposure, hospitalisation or travel history in the last 3 months were plated on screening agar. Enterobacterales were identified by MALDI-TOF MS and antibiotic susceptibility testing and whole-genome sequencing performed.

The paper is well written and all the results are clearly presented. It presents valuable information on the prevalence of community CPE in Pakistan. WGS highlights the dominant sequence types carrying CPE genes and provides epidemiological information on plasmids carrying carbapenemase genes.

I have the following comments and suggestions for the authors:

METHODS:

Page 5, line 98: Antimicrobial susceptibility testing. It would add to the manuscript if the authors could perform susceptibility testing against newer agents, like aztreonam+avibactam and cefiderocol.

Response: the association aztreonam+avibactam has been tested for a subset of CPE (NDM-5 producing E. coli), the results being reported in this article: "International circulation of aztreonam/avibactam-resistant NDM-5-producing Escherichia coli isolates: successful epidemic clones." Journal of global antimicrobial resistance 27 (2021): 326-328. Indeed, a significant number of strains combining NDM-5, CMY-42 and specific insertions motives in the PBP3 showed a decreased susceptibility to aztreonam-avibactam. 

As for cefidérocol, it is not available in Pakistan. In France, we currently have a shortage for reagents required to perform microdilution tests (E-tests strips and discs being not recommended for céfidérocol testing). Nonetheless, we keep this in mind for our future work. 

RESULTS:

Sampling: Page 8, line 160-162: In this study a young patient cohort was investigated (average age 38 and 44 years). How does this compare to other studies from the region? Could this influence the finding?

Response: We have added a paragraph regarding influence of young patient cohort on the findings in discussion part, also reference of few studies showing children and older people are at high risk of acquisition. We agree that the prevalence of CPE could be higher in at-risk populations such as elderly and young children. Page no. 15, L: 326-332.

Prevalence of CPE carriage: Page 8, line 169: It would be interesting to add some information on whether the high community prevalence was reflected in disease burden in the hospital. How many of the patients required treatment for CPE? Were any of the hospital acquired cases considered to be linked?

Response: Information has been added regarding situation of patients from which samples were collected on Page no. 9 (lines 181-182). Patients were admitted for or pregnancy-related planned consultation. We did not record the occurrence of infections during the follow-up, but given our results, such study would be interesting to implement. As for the connection between community and hospital-acquired strains, we only screened for hospital-acquired CPE in patients found negative at admission. To see whether community-acquired strains can cause hospital-onset infections, we should consider CPE-positive patients at admission. Of note, few community and hospital acquired cases were closely linked as mentioned on Page no. 11, lines 233-235.

Phylogeny: Page 10, line 220: The authors comment on the maximal genetic distances but do not discuss genetic similarities. On the tree some of the sequences look like they are related? Did they find possible relatedness between community and hospital acquired strains indicating possible cross transmission? Could the authors comment on this?

Response: This has been mentioned in the results section, Phylogeny at Page 11 (lines 234 – 237).

Relative abundance: Page 11, line 239: The authors present data on quantification of CRE but do not discuss the finding. Did they expect a difference in the relative abundance between community-acquired CRE and hospital-associated CRE? Does this depend on whether the patient receives antibiotics? I suggest to briefly mention this in the discussion.

Response: We have discussed these points in the discussion section on page No. 15-16 (lines 333-337).

DISCUSSION:

Line 248: ‘CPE carriage is mostly associated with hospital environment’ Did the other studies undertaken in Pakistan focus on hospital acquired cases? It would be interesting to discuss isolates that were acquired in hospital here. On the tree they look spread out over the clades but closely related to some of the community isolates? What is known about hospital acquisition of CPE in hospitals in Pakistan? How many of the CPE positive patients did require CPE treatment? What impact does the high community carriage rate have on hospitals?

Response: Limited data is available regarding community carriage of CPE in Pakistan and mostly studies are based on hospital isolates. Hospital acquired isolates have been mentioned in discussion, along with their relatedness with community associated isolates. Factors which are responsible for CPE acquisition in hospitals and impact high community carriage on hospitals have also been included in revised version. (please see Page No. 12-13, lines 260-269). Data about CPE positive patients requiring CPE treatment was not collected. The impact of high community carriage has been discussed on pages 12-13 (lines 260 – 269). 

Line 278: ‘Combination of resistance genes, virulence, and ability to easily disseminate…’ The authors mention virulence here but do not discuss virulence data for the observed ST types. How frequent is ST 167 found in clinical samples?

Response: The word virulence has been removed from the description of ST 167. The prevalence and importance of ST 167 in clinical settings has been discussed in detail at page 14 (lines 298 – 300). 

Line 295/296: It would add to the value of the manuscript to be able to discuss cefiderocol susceptibilities here. It might not be available as an agent in Pakistan yet but with the high prevalence of metallo-b-lactamases it would be helpful to assess its activity against the isolates of the study.

Response: Following the Reviewer 2’s comment on antibiotics potentially active on CPE, cefiderocol is not available in Pakistan as a drug or for testing. In France, we currently have a shortage for reagents required to perform microdilution tests (E-tests strips and discs being not recommended for céfidérocol testing). Nonetheless, we keep this in mind for our future work. 

Besides, the association aztreonam+avibactam has been tested for a subset of CPE (NDM-5 producing E. coli), the results being reported in this article: "International circulation of aztreonam/avibactam-resistant NDM-5-producing Escherichia coli isolates: successful epidemic clones." Journal of global antimicrobial resistance 27 (2021): 326-328. Indeed, a significant number of strains combining NDM-5, CMY-42 and specific insertions motives in the PBP3 showed a decreased susceptibility to aztreonam-avibactam.

---

## [Decision Letter · Decision Letter 1]

7 Feb 2022

PONE-D-21-36109R1Dissemination of Carbapenemase-producing Enterobacterales in the community of Rawalpindi, PakistanPLOS ONE

Dear Dr. Zahra,

Thank you for submitting your manuscript to PLOS ONE. After careful consideration, we feel that it has merit but does not fully meet PLOS ONE’s publication criteria as it currently stands. Therefore, we invite you to submit a revised version of the manuscript that addresses the points raised during the review process.

If applicable, we recommend that you deposit your laboratory protocols in protocols.io to enhance the reproducibility of your results. Protocols.io assigns your protocol its own identifier (DOI) so that it can be cited independently in the future. For instructions see: https://journals.plos.org/plosone/s/submission-guidelines#loc-laboratory-protocols. Additionally, PLOS ONE offers an option for publishing peer-reviewed Lab Protocol articles, which describe protocols hosted on protocols.io. Read more information on sharing protocols at https://plos.org/protocols?utm_medium=editorial-emailutm_source=authorlettersutm_campaign=protocols.

We look forward to receiving your revised manuscript.

Kind regards,

Zhi Ruan, Ph.D.

Academic Editor

PLOS ONE

Journal Requirements:

Reviewers' comments:

Reviewer's Responses to Questions

**Comments to the Author**

1. If the authors have adequately addressed your comments raised in a previous round of review and you feel that this manuscript is now acceptable for publication, you may indicate that here to bypass the “Comments to the Author” section, enter your conflict of interest statement in the “Confidential to Editor” section, and submit your "Accept" recommendation.

Reviewer #1: All comments have been addressed

Reviewer #2: (No Response)

Reviewer #3: All comments have been addressed

2. Is the manuscript technically sound, and do the data support the conclusions?

Reviewer #1: Yes

Reviewer #2: Yes

Reviewer #3: (No Response)

3. Has the statistical analysis been performed appropriately and rigorously? 

Reviewer #1: I Don't Know

Reviewer #2: No

Reviewer #3: (No Response)

4. Have the authors made all data underlying the findings in their manuscript fully available?

Reviewer #1: Yes

Reviewer #2: Yes

Reviewer #3: (No Response)

5. Is the manuscript presented in an intelligible fashion and written in standard English?

Reviewer #1: Yes

Reviewer #2: Yes

Reviewer #3: (No Response)

6. Review Comments to the Author

Reviewer #1: There is incorrect on lines 327-328 "A study from Thailand also reported similar findings of mcr-1 detection from asymptomatic human feces in 2012 [52]." That study is conducted on patients with variety specimens, no feces. The author should be check and carefully mentioned. Please revise this sentence again.

Reviewer #2: Overall, it has improved, but there was one point that was not answered satisfactorily because the previous my review was unclear.

L. 150-158 Methods, Quantification of CRE

The counting method regarding the total amount of Gram-negative bacteria is unclear. According to the description in the text, rectal specimens were first inoculated onto ertapenem-supplemented agar plates. Therefore, even if the specimens were cultured on agar plates without ertapenem, the total amount of Gram-negative bacteria may not have been counted accurately, and those selected for ertapenem may have been counted. In such a case, the number of Gram-negative bacteria would be low and the CRE would be overestimated, which raises questions about the reliability of the CRE-RA and is unacceptable. The method of counting total Gram-negative bacteria should be clearly explained.

Reviewer #3: (No Response)

7. PLOS authors have the option to publish the peer review history of their article (what does this mean?). If published, this will include your full peer review and any attached files.

Reviewer #1: No

Reviewer #2: **Yes: **MASAHIRO SUZUKI

Reviewer #3: No

---

## [Author Response · Author response to Decision Letter 1]

17 Feb 2022

Reviewer's comments:

Reviewer #1: There is incorrect on lines 327-328 "A study from Thailand also reported similar findings of mcr-1 detection from asymptomatic human feces in 2012 [52]." That study is conducted on patients with variety specimens, no feces. The author should be check and carefully mentioned. Please revise this sentence again.

Response: Thank you for pointing out this mistake. Correction has been made. Page 15, Line, 329.

Reviewer #2:

Overall, it has improved, but there was one point that was not answered satisfactorily because the previous my review was unclear.

L. 150-158 Methods, Quantification of CRE

The counting method regarding the total amount of Gram-negative bacteria is unclear. According to the description in the text, rectal specimens were first inoculated onto ertapenem-supplemented agar plates. Therefore, even if the specimens were cultured on agar plates without ertapenem, the total amount of Gram-negative bacteria may not have been counted accurately, and those selected for ertapenem may have been counted. In such a case, the number of Gram-negative bacteria would be low and the CRE would be overestimated, which raises questions about the reliability of the CRE-RA and is unacceptable. The method of counting total Gram-negative bacteria should be clearly explained.

Response: Thank you for pointing out the confusion in methods. The rectal swabs were not directly plated on the plates. Instead they were thoroughly mixed in normal saline first and then 100 ul of the swab saline mix was plated on MacConkey agar with and without antibiotic directly and serial dilutions were also made to calculate relative abundance. This methodology was followed based on previously published methods in following articles: ‘Relative Fecal Abundance of Extended-Spectrum-β-Lactamase-Producing Escherichia coli Strains and Their Occurrence in Urinary Tract Infections in Women. Antimicrob. Agents Chemother. 57, 4512–4517 (2013)’ and ‘High Rate of Acquisition but Short Duration of Carriage of Multidrug-Resistant Enterobacteriaceae After Travel to the Tropics. Clin. Infect. Dis. 61, 593–600 (2015)’

The methods part has been re-phrased to make it more clear. Please see page No. 5, line 98 and page No. 7, lines 151-153.

---

## [Decision Letter · Decision Letter 2]

2 Mar 2022

PONE-D-21-36109R2Dissemination of Carbapenemase-producing Enterobacterales in the community of Rawalpindi, PakistanPLOS ONE

Dear Dr. Zahra,

Thank you for submitting your manuscript to PLOS ONE. After careful consideration, we feel that it has merit but does not fully meet PLOS ONE’s publication criteria as it currently stands. Therefore, we invite you to submit a revised version of the manuscript that addresses the points raised during the review process.

ACADEMIC EDITOR: A major revision is required.Please revise the manuscript according to the reviewer comments./==============================

If applicable, we recommend that you deposit your laboratory protocols in protocols.io to enhance the reproducibility of your results. Protocols.io assigns your protocol its own identifier (DOI) so that it can be cited independently in the future. For instructions see: https://journals.plos.org/plosone/s/submission-guidelines#loc-laboratory-protocols. Additionally, PLOS ONE offers an option for publishing peer-reviewed Lab Protocol articles, which describe protocols hosted on protocols.io. Read more information on sharing protocols at https://plos.org/protocols?utm_medium=editorial-emailutm_source=authorlettersutm_campaign=protocols.

We look forward to receiving your revised manuscript.

Kind regards,

Abdelazeem Mohamed Algammal, Prof, Ph.D

Academic Editor

PLOS ONE

Reviewers' comments:

Reviewer's Responses to Questions

**Comments to the Author**

1. If the authors have adequately addressed your comments raised in a previous round of review and you feel that this manuscript is now acceptable for publication, you may indicate that here to bypass the “Comments to the Author” section, enter your conflict of interest statement in the “Confidential to Editor” section, and submit your "Accept" recommendation.

Reviewer #4: (No Response)

2. Is the manuscript technically sound, and do the data support the conclusions?

Reviewer #4: Partly

3. Has the statistical analysis been performed appropriately and rigorously? 

Reviewer #4: Yes

4. Have the authors made all data underlying the findings in their manuscript fully available?

Reviewer #4: Yes

5. Is the manuscript presented in an intelligible fashion and written in standard English?

Reviewer #4: Yes

6. Review Comments to the Author

Reviewer #4: Comments to authors:

-The current study is interesting; however, the authors should address the following comments to improve the quality of the manuscript:

- Please write the scientific names of bacterial pathogens and genes in the correct form all over the manuscript and the references section.

Title:

I think the work would benefit from the title that contains the main conclusion of the study (should be derived from the conclusion). Please modify the title.

Abstract:

- The abstract must illustrate the used methods and the most prevalent results (give more hints about methods and results). Besides, rephrase the aim of the work and the main conclusion of your findings.

-Introduction: (it needs to be more informative):

-Give a hint about the virulence factors, different infections caused by E.coli and K. pneumoniae, and the mechanism of disease occurrence.

- The authors should illustrate the public health importance concerning the emergence of multidrug-resistant (MDR) bacterial pathogens that reflect the necessity of new potent and safe antimicrobial agents. Several studies proved the widespread MDR- bacterial pathogens;

Authors could add the following paragraph:

Multidrug resistance has been increased all over the world that is considered a public health threat. Several recent investigations reported the emergence of multidrug-resistant bacterial pathogens from different origins including humans, birds, cattle, and fish that increase the necessity of new potent and safe antimicrobial agents. Besides, the routine application of the antimicrobial susceptibility testing to detect the antibiotic of choice as well as the screening of the emerging MDR strains. You are advised to cite the following valuable studies:

1.PMID: 33177849

2.PMID: 33188216

3.PMID: 30150182

4.PMID: 33947875

5.PMID: 32994450

6. PMID: 32497922

7.PMID: 33061472

8.PMID: 34445951

9.https://doi.org/10.5114/ceji.2013.3774020 .

10.https://doi.org/10.1016/j.aquaculture.2021.737643

11.https://doi.org/10.1016/j.foodcont.2021.108066

-Rephrase the aim of the work to be clear and better sound.

Material and methods:

-Add this subtitle: Bacterial Isolation and identification:

•Discuss in detail the methods of isolation and identification of Enterobacterales, especially, E. coli and K. pneumoniae. Besides, specific references should be added.

•Add the company, city, and country of the used bacterial media and reagents that were used in the biochemical identification of isolates. Also, enumerate all used biochemical reactions.

- Modify the subtitile to be: Antimicrobial susceptibility testing:

-Illustrate the antimicrobial classes of the tested antibiotics.

-The authors are advised to classify the tested isolates to MDR , XDR, and PDR as described by Magiorakos et al.

Magiorakos AP, Srinivasan A, Carey RB, Carmeli Y, Falagas ME, Giske CG, et al. Multidrug-resistant, extensively drug-resistant and pandrug-resistant bacteria: An international expert proposal for interim standard definitions for acquired resistance. Clin Microbiol Infect. 2012; 18:268–81. doi:10.1111/j.1469-0691.2011.03570.x.

- The correlation between phenotypic and genotypic multidrug resistance should be performed.

-Add the following subtitle to the methods section, and disscuss the method in detail ( Support the methods with specific references):

Multilocus sequence typing (MLST)

-Statistical analyses:

-Add more details about the software used in the statistical analyses.

-Results:

-Add this subtitle: Phenotypic characteristics of the recovered isolates:

•Illustrate in detail the phenotypic characteristics of the recovered E. coli and K. pneumoniae isolates.

- Illustrate in a table the results of The Antimicrobial susceptibility testing.

•Illustrate in a new table the occurrence of MDR (Multidrug resistance) among the recovered isolates as the following (illustrate the names of the antimicrobial classes and different antibiotics):

No. of strains%Type of resistance

R, MDR, and XDRPhenotypic multidrug resistance

(Antimicrobial classes and different antibiotics).The antibiotic -resistance genes

-The correlation (Correlation coefficient) between phenotypic and genotypic multidrug resistance should be performed.

- Add the supplemetary Figures to the main manuscript.

-Increase the resution of all Figures ( it should be 600 dpi).

-Discussion:

- The authors are advised to illustrate the real impact of their findings without repetition of results.

-Illustrate the different mechanisms of antimicrobial resistance in E. coli and K. pneumoniae.

-Conclusion

- Should be rephrased to be consise and sounded. A real conclusion should focus on the question or claim you articulated in your study, which resolution has been the main objective of your paper?

7. PLOS authors have the option to publish the peer review history of their article (what does this mean?). If published, this will include your full peer review and any attached files.

Reviewer #4: No

---

## [Author Response · Author response to Decision Letter 2]

15 Apr 2022

Reviewer #4: Comments to authors:

The current study is interesting; however, the authors should address the following comments to improve the quality of the manuscript:

Please write the scientific names of bacterial pathogens and genes in the correct form all over the manuscript and the references section.

Response: Bacterial pathogens and genes names have been re-checked and corrected where needed, Page No. 2, lines 43-44 and Page No. 21-27, lines 459, 462, 466, 472, 521, 529, 540, 583, 626, 630, 633, 643,646, 650, 653, 660, 664, 668, 672, 679, 683, 688.

Title:

I think the work would benefit from the title that contains the main conclusion of the study (should be derived from the conclusion). Please modify the title.

Response: This manuscript has been reviewed two times previously and we already made changes based on previous reviews. Furthermore, we found dissemination of many clones of bacteria, not just one specific; hence we are keeping a generalized title. 

Abstract:

The abstract must illustrate the used methods and the most prevalent results (give more hints about methods and results). Besides, rephrase the aim of the work and the main conclusion of your findings.

Response: Methods have been added, Page No. 2, lines: 38-42 while most prevalent results have already been described in abstract. Please see Page No. 2, lines 43-49 and this was reviewed two times previously and we had changed it based on previous reviews. The main conclusion is re-phrased; Pages No. 2-3, lines: 49-53. 

Introduction: (it needs to be more informative):

Give a hint about the virulence factors, different infections caused by E. coli and K. pneumoniae, and the mechanism of disease occurrence.

Response: Virulence factors, infections, and mechanism of disease occurrence have been added in introduction to make it more informative. Please see Page No. 3, lines 57-64.

The authors should illustrate the public health importance concerning the emergence of multidrug-resistant (MDR) bacterial pathogens that reflect the necessity of new potent and safe antimicrobial agents. Several studies proved the widespread MDR- bacterial pathogens;

Authors could add the following paragraph:

Multidrug resistance has been increased all over the world that is considered a public health threat. Several recent investigations reported the emergence of multidrug-resistant bacterial pathogens from different origins including humans, birds, cattle, and fish that increase the necessity of new potent and safe antimicrobial agents. Besides, the routine application of the antimicrobial susceptibility testing to detect the antibiotic of choice as well as the screening of the emerging MDR strains. You are advised to cite the following valuable studies:

1. PMID: 33177849 

2. PMID: 33188216

3. PMID: 30150182 

4. PMID: 33947875 

5. PMID: 32994450

6. PMID: 32497922 

7. PMID: 33061472

8. PMID: 34445951

9. https://doi.org/10.5114/ceji.2013.3774020 .

10. https://doi.org/10.1016/j.aquaculture.2021.737643

11 https://doi.org/10.1016/j.foodcont.2021.108066

Response: The suggested paragraph is a generalized information on AMR which is suitable for a study focusing on AMR and One health. Our manuscript is focused work investigating the carriage of CREs and CPEs in community and the related information on Enterbacterales and carbapenem resistance mechanisms have been described in the introduction. We have added information on virulence, and diseases as suggested in prior comments. We appreciated the advice to add valuable studies but the references advised above are not significantly relevant to our work and do not add any value to the work, hence these have not been added. 

Rephrase the aim of the work to be clear and better sound.

Response: Aim of the work has been rephrased, Page No.4, lines 88-92. 

Material and methods:

Add this subtitle: Bacterial Isolation and identification 

Response: Subtitle and relevant detail has been added. Page No. 5, line no. 104. 

Discuss in detail the methods of isolation and identification of Enterobacterales, especially, E. coli and K. pneumoniae. Besides, specific references should be added. 

Response: The methods of isolation have already been discussed, Page No.5, line no. 112-115 while identification methods of Enterobacterales with references have been added, Page No. 6 line no. 115-126. 

Add the company, city, and country of the used bacterial media and reagents that were used in the biochemical identification of isolates. Also, enumerate all used biochemical reactions.

Response: Biochemical reactions, company, city, and country of all used bacterial media and reagents have been added; Page No. 6, line no. 118-125.

Modify the subtitle to be: Antimicrobial susceptibility testing: 

Illustrate the antimicrobial classes of the tested antibiotics

Response: Subtitle named antimicrobial susceptibility testing has been added, Page No. 6, line no. 128, and antimicrobial classes have been added to the material and method section under the heading of antimicrobial susceptibility testing; pages 6-7, line no. 130-141. 

The authors are advised to classify the tested isolates to MDR, XDR, and PDR as described by Magiorakos et al.

Magiorakos AP, Srinivasan A, Carey RB, Carmeli Y, Falagas ME, Giske CG, et al. Multidrug-resistant, extensively drug-resistant and pandrug-resistant bacteria: An international expert proposal for interim standard definitions for acquired resistance. Clin Microbiol Infect. 2012; 18:268–81. doi:10.1111/j.1469-0691.2011.03570. x. 

Response:

Response: Categories of isolates have been made to XDR, and MDR as described by Magiorakos et al., Page No.12, lines 247-250. No isolate was found to be PDR. 

The correlation between phenotypic and genotypic multidrug resistance should be performed

Response: The work aimed to assess the carriage of CREs and CPEs in community. Antibiotic susceptibility were analyzed for all isolates (n=170) while the detailed genomic analysis was done for 78 isolates after de-duplication based on antibiotic susceptibility profiles. The assessment of methodologies in detecting the resistance was not the scope of the work hence the correlation of phenotypic and genotypic methods has not been carried out in this work. 

Add the following subtitle to the methods section and discuss the method in detail (Support the methods with specific references):

Multilocus sequence typing (MLST), Statistical analyses:

Add more details about the software used in the statistical analyses.

Response: Subtitles and relevant detail of MLST with references (Page No. 7, lines 155-158) and Statistical analyses along with references, and detail of software has been added; Page No. 9 line no. 192-197. 

Results:

Add this subtitle: Phenotypic characteristics of the recovered isolates: Illustrate in detail the phenotypic characteristics of the recovered E. coli and K. pneumoniae isolates.

Response: Subtitle and characteristics of recovered Enterobacterales isolates have been added, Page No. 11, line no. 221-224. 

. 

Illustrate in a table the results of The Antimicrobial susceptibility testing

Table of antimicrobial susceptibility testing results has been added in supporting material as S1 Table (Page No. 12, line No. 243 and Page No. 28, line 690. 

Illustrate in a new table the occurrence of MDR (Multidrug resistance) among the recovered isolates as the following (illustrate the names of the antimicrobial classes and different antibiotics): 

No. of strains%, Type of resistance

R, MDR, and XDR Phenotypic multidrug resistance

(Antimicrobial classes and different antibiotics). The antibiotic -resistance genes

Response: Table showing Isolates categorization into XDR and MDR has been added in the supporting material (S2 Table: Page No. 28, line 691), showing number of strains, %, type of resistance and phenotypic multidrug resistance and susceptibility (Antimicrobial classes and different antibiotics), while detail of all antibiotic resistance genes has already been described in (S3 Table: Page No. 28, line no. 692).

The correlation (Correlation coefficient) between phenotypic and genotypic multidrug resistance should be performed. 

Response: The work aimed to assess the carriage of CREs and CPEs in community. Antibiotic susceptibility were analyzed for all isolates (n=170) while the detailed genomic analysis was done for 78 isolates after de-duplication based on antibiotic susceptibility profiles. The assessment of methodologies in detecting the resistance was not the scope of the work hence the correlation of phenotypic and genotypic methods has not been carried out in this work. 

Add the supplementary Figures to the main manuscript.

Response: All Supplementary figures have been added to main manuscript, Page No. 10, lines 215-217, Page No. 13, lines. 278-283, Page No. 15, lines.318-319, Page No. 16, line no. 326-328.

Increase the resolution of all Figures (it should be 600 dpi).

Response: Resolution of all figures has been increased to 600 dpi. 

Discussion:

The authors are advised to illustrate the real impact of their findings without repetition of results.

Response: Impact of study findings has been added to discussion, Page No.20, Line no. 413-417.

Illustrate the different mechanisms of antimicrobial resistance in E. coli and K. pneumoniae.

Response: Prevalent resistance mechanisms in CRE have already been discussed in Introduction, Pages 3-4, line no. 71-80.

Conclusion

Should be rephrased to be concise and sounded. A real conclusion should focus on the question or claim you articulated in your study, which resolution has been the main objective of your paper?

Response: Conclusion has been improved according to suggestion, Pages 20-21, lines 427-435.

---

## [Decision Letter · Decision Letter 3]

13 May 2022

PONE-D-21-36109R3Dissemination of Carbapenemase-producing Enterobacterales in the community of Rawalpindi, PakistanPLOS ONE

Dear Dr. Zahra,

Thank you for submitting your manuscript to PLOS ONE. After careful consideration, we feel that it has merit but does not fully meet PLOS ONE’s publication criteria as it currently stands. Therefore, we invite you to submit a revised version of the manuscript that addresses the points raised during the review process.

Specifically, please provide your responses to the reviewer's remaining concerns.

We look forward to receiving your revised manuscript.

Kind regards,

Jianhong Zhou

Staff Editor

PLOS ONE

Journal Requirements:

Reviewers' comments:

Reviewer's Responses to Questions

**Comments to the Author**

1. If the authors have adequately addressed your comments raised in a previous round of review and you feel that this manuscript is now acceptable for publication, you may indicate that here to bypass the “Comments to the Author” section, enter your conflict of interest statement in the “Confidential to Editor” section, and submit your "Accept" recommendation.

Reviewer #1: All comments have been addressed

Reviewer #2: (No Response)

Reviewer #3: All comments have been addressed

2. Is the manuscript technically sound, and do the data support the conclusions?

Reviewer #1: Yes

Reviewer #2: Yes

Reviewer #3: Yes

3. Has the statistical analysis been performed appropriately and rigorously? 

Reviewer #1: N/A

Reviewer #2: Yes

Reviewer #3: N/A

4. Have the authors made all data underlying the findings in their manuscript fully available?

Reviewer #1: No

Reviewer #2: (No Response)

Reviewer #3: Yes

5. Is the manuscript presented in an intelligible fashion and written in standard English?

Reviewer #1: Yes

Reviewer #2: Yes

Reviewer #3: Yes

6. Review Comments to the Author

Reviewer #1: This revised manuscript has been address all points of the reviewer's comments. I satisfy the response of the authors.

Reviewer #2: My concerns previously noted have been corrected, and together with the corrections indicated by Reviewer 4, the manuscript is much better. I have commented on the parts of the text that the author refused in response to Reviewer 4's remarks as follows.

The comment refused by authors: Title

Comments: No modification is required. I do not believe that the title change is necessary because, as the authors claim, the study investigated the genus Enterobacterales bacteria.

The comment refused by authors: References assigned by Reviewer 4

Comments: No modification is required. The references assigned by Reviewer 4 are biased in authorship and are not impartial. I believe they should NOT be added.

The comment refused by authors: Discuss in detail the methods of isolation and identification of Enterobacterales, especially, E. coli and K. pneumoniae.

Comments: Modification is required. Selection plates and concentration of antimicrobials used will affect the detection rate. It is a good idea to add this information to the “limitation” in discussion.

The comment refused by authors: The correlation between phenotypic and genotypic multidrug resistance should be performed

Comments: Modification is required. I think, Reviewer 4 may not have pointed out the methodologies. The paper does include antimicrobial susceptibility tests and a genetic study, which suggests that the necessary experiments have been performed. However, it does not present data linking the phenotypic antimicrobial susceptibility to the genotype. In particular, a number of strains have been found to be susceptible to monobactam, aminoglycosides, and imipenem, and the correlation with genotype needs to be discussed.

The comment refused by authors: Illustrate the different mechanisms of antimicrobial resistance in E. coli and K. pneumoniae.

Comments: No modification is required. As the author claims, the drug resistance mechanism is introduced in the introduction, so there is no need to add it to the discussion again.

Reviewer #3: (No Response)

7. PLOS authors have the option to publish the peer review history of their article (what does this mean?). If published, this will include your full peer review and any attached files.

Reviewer #1: No

Reviewer #2: **Yes: **Masahiro Suzuki

Reviewer #3: No

---

## [Author Response · Author response to Decision Letter 3]

20 May 2022

RE: PLOS One (PONE-D-21-36109R3)

Title: Dissemination of Carbapenemase-producing Enterobacterales in the community of Rawalpindi, Pakistan

Review Comments to the Author

Reviewer #1: This revised manuscript has been addressing all points of the reviewer's comments. I satisfy the response of the authors.

Reviewer #2: My concerns previously noted have been corrected, and together with the corrections indicated by Reviewer 4, the manuscript is much better. I have commented on the parts of the text that the author refused in response to Reviewer 4's remarks as follows.

The comment refused by authors: Title

Comments: No modification is required. I do not believe that the title change is necessary because, as the authors claim, the study investigated the genus Enterobacterales bacteria.

The comment refused by authors: References assigned by Reviewer 4

Comments: No modification is required. The references assigned by Reviewer 4 are biased in authorship and are not impartial. I believe they should NOT be added.

The comment refused by authors: Discuss in detail the methods of isolation and identification of Enterobacterales, especially, E. coli and K. pneumoniae.

Comments: Modification is required. Selection plates and concentration of antimicrobials used will affect the detection rate. It is a good idea to add this information to the “limitation” in discussion.

Response: The methods of isolation and identification of Enterobacterales have already been discussed page No. 5-6, lines 102-125. The limitation of selection plates and concentration of antibiotics has been added to the limitation of study on Pages 19-20. lines 415-418. 

The comment refused by authors: The correlation between phenotypic and genotypic multidrug resistance should be performed

Comments: Modification is required. I think, Reviewer 4 may not have pointed out the methodologies. The paper does include antimicrobial susceptibility tests and a genetic study, which suggests that the necessary experiments have been performed. However, it does not present data linking the phenotypic antimicrobial susceptibility to the genotype. In particular, a number of strains have been found to be susceptible to monobactam, aminoglycosides, and imipenem, and the correlation with genotype needs to be discussed.

Response: The data of the observed phenotypic resistance along with the genomic content of the antibiotic resistance genes has been added. Since this information is not in the scope of main findings of the study, this has been included in the supplementary information as S4 Table. 

The comment refused by authors: Illustrate the different mechanisms of antimicrobial resistance in E. coli and K. pneumoniae.

Comments: No modification is required. As the author claims, the drug resistance mechanism is introduced in the introduction, so there is no need to add it to the discussion again.

Reviewer #3: (No Response)

---

## [Decision Letter · Decision Letter 4]

16 Jun 2022

Dissemination of Carbapenemase-producing Enterobacterales in the community of Rawalpindi, Pakistan

PONE-D-21-36109R4

Dear Dr. Zahra,

We’re pleased to inform you that your manuscript has been judged scientifically suitable for publication and will be formally accepted for publication once it meets all outstanding technical requirements.

Kind regards,

Jianhong Zhou

Staff Editor

PLOS ONE

Additional Staff Editor Comments: Please update your data availability statement on the submission details page by including the accession number PRJNA645311 as this will be used for the Data Availability Statement in the published article.  statementonthesubmissiondetailspageasthiswillbeusedfortheDASinthepublishedarticle.

Reviewers' comments:

Reviewer's Responses to Questions

**Comments to the Author**

1. If the authors have adequately addressed your comments raised in a previous round of review and you feel that this manuscript is now acceptable for publication, you may indicate that here to bypass the “Comments to the Author” section, enter your conflict of interest statement in the “Confidential to Editor” section, and submit your "Accept" recommendation.

Reviewer #2: All comments have been addressed

2. Is the manuscript technically sound, and do the data support the conclusions?

Reviewer #2: Yes

3. Has the statistical analysis been performed appropriately and rigorously? 

Reviewer #2: Yes

4. Have the authors made all data underlying the findings in their manuscript fully available?

Reviewer #2: Yes

5. Is the manuscript presented in an intelligible fashion and written in standard English?

Reviewer #2: Yes

6. Review Comments to the Author

Reviewer #2: (No Response)

7. PLOS authors have the option to publish the peer review history of their article (what does this mean?). If published, this will include your full peer review and any attached files.

Reviewer #2: **Yes: **Masahiro Suzuki

---

## [Editor Report · Acceptance letter]

30 Jun 2022

PONE-D-21-36109R4 

Dissemination of Carbapenemase-producing Enterobacterales in the community of Rawalpindi, Pakistan 

Dear Dr. Zahra:

I'm pleased to inform you that your manuscript has been deemed suitable for publication in PLOS ONE. Congratulations! Your manuscript is now with our production department. 

Kind regards, 

on behalf of

Jianhong Zhou 

Staff Editor

PLOS ONE